# Transmission-clearance trade-offs indicate that dengue virulence evolution depends on epidemiological context

Rotem Ben-Shachar[1,2,3,4] & Katia Koelle[2,5]

An extensive body of theory addresses the topic of pathogen virulence evolution, yet few studies have empirically demonstrated the presence of fitness trade-offs that would select for intermediate virulence. Here we show the presence of transmission-clearance trade-offs in dengue virus using viremia measurements. By fitting a within-host model to these data, we further find that the interaction between dengue and the host immune response can account for the observed trade-offs. Finally, we consider dengue virulence evolution when selection acts on the virus's production rate. By combining within-host model simulations with empirical findings on how host viral load affects human-to-mosquito transmission success, we show that the virus's transmission potential is maximized at production rates associated with intermediate virulence and that the optimal production rate critically depends on dengue's epidemiological context. These results indicate that long-term changes in dengue's global distribution impact the invasion and spread of virulent dengue virus genotypes.

[1] Program in Computational Biology and Bioinformatics, Duke University, Durham, NC 27708, USA. [2] Department of Biology, Duke University, Durham, NC 27708, USA. [3] Department of Integrative Biology, University of California, Berkeley, CA 94720, USA. [4] Division of Infectious Diseases and Vaccinology, School of Public Health, University of California, Berkeley, CA 94720, USA. [5] Department of Biology, Emory University, Atlanta, GA 30322, USA. Correspondence and requests for materials should be addressed to R.B.-S. (email: rbenshachar@gmail.com) or to K.K. (email: katia.koelle@emory.edu)

Evolution of virulence theory proposes that parasites will evolve to an intermediate level of virulence when a trade-off exists between parasite transmissibility and virulence, where virulence is most commonly defined as the rate at which hosts experience disease-induced mortality[1,2]. Although a large body of literature contributes to virulence theory, few studies have empirically demonstrated the existence of a fitness trade-off that would result in evolution towards intermediate virulence[2]. Among human pathogens, HIV provides a rare, well-documented example[3,4]. Human malaria provides another example, where observed, broad patterns between virulence and transmissibility are consistent with evolution of virulence theory[5]. While a trade-off between transmission rate and disease-induced mortality is the classical trade-off considered in evolution of virulence theory, early work has also considered alternative trade-offs, for example, between the rate of recovery and the rate of disease-induced mortality[6].

More generally, virulence can be defined as the ability of a pathogen to cause disease[7]. In this case, evolution towards intermediate virulence can also result from trade-offs in many different fitness components[8], rather than only through the classical trade-off between transmission and disease-induced host mortality. Indeed, recent studies have indicated that the pathogen clearance rate is likely to be an important fitness component to consider in the evolution of virulence[8,9], defined as disease severity. These studies further underscore that the interaction between pathogens and the host immune response may be an important factor in the evolution of pathogen virulence[8,10]. Here, with dengue virus, we document an example of a human pathogen subject to a fitness trade-off involving viral clearance.

In brief, dengue is a vector-borne virus that infects up to 400 million individuals annually[11]. Dengue infections can be asymptomatic, result in symptomatic dengue fever (DF) or result in severe disease, defined as dengue hemorrhagic fever (DHF) or dengue shock syndrome (DSS)[12]. The virus comprises four antigenically distinct serotypes, each of which is genetically structured into several clades called genotypes, based on levels of nucleotide divergence exceeding 6%[13]. Dengue virus infection induces long-lived immunity against the infecting serotype[14], but only a transient period of cross-protection against heterologous dengue serotypes[14–16]. Primary dengue infections can result in severe disease, but they do so rarely, with the majority of these infections being asymptomatic or resulting in DF[14]. In contrast, secondary heterologous infections have an appreciably higher risk of developing into severe disease[14]. Tertiary and quaternary (henceforth, "post-secondary") infections contribute very little to dengue hospital admissions[14,17], although they are known to occur. The elevated risk of developing severe disease with a heterologous secondary infection is due at least in part to antibody-dependent enhancement (ADE), a process by which antibodies generated during a primary infection facilitate viral entry into host target cells bearing Fc−γ receptors during a secondary infection[14,18]. This process, among others, results in an elevated number of cytokine-secreting infected cells, which can initiate a cytokine storm that ultimately leads to the development of vascular leakage[14,18–20], the hallmark of DHF. Severe dengue disease is therefore a result of immunopathology, whereby the host's own immune response, rather than the presence of the pathogen, is the direct cause of damage to host tissue.

Using viremia measurements from symptomatic dengue-infected patients, we first document an empirical trade-off between peak viral load and the viral clearance rate, indicating that high viral loads come at the cost of accelerated viral clearance in both primary and secondary dengue infections. By fitting an existing within-host dengue model to the observed viremia measurements, we show that this observed trade-off can be explained by the interaction of the virus with the host immune response, consistent with recent arguments that the host immune response can be an important factor limiting the evolution of higher transmission rates[10,21].

Given that the probability of successfully transmitting to dengue's mosquito vector strongly depends on host viral load[22], and that high peak viral load is a strong predictor of the probability of triggering severe dengue disease[23–26], the empirical trade-off we document between peak viral load and viral clearance rate sets the stage for the possibility that dengue virus strains associated with intermediate virulence maximize the virus's transmission potential. We consider how selection may act on a within-host viral phenotype—the viral production rate—and the implications of this selection on dengue virulence evolution. We show that dengue virus transmission potential is maximized at production rates that are associated with intermediate peak viral loads, and thus an intermediate risk of triggering severe disease. Our analysis further shows that optimal virulence, defined as disease severity, depends on epidemiological context. Specifically, we show that even though secondary infections cause more severe disease than primary infections, they may place selection pressures on the virus to evolve lower virulence in any given infection. These results indicate that the changing global landscape of dengue endemism[27,28] may in the long-term affect not only infection levels, but, via evolution, disease risk.

## Results

**Trade-off between peak viral load and viral clearance rate.** In the context of dengue, virulence can be defined as the probability that infection results in severe disease, defined classically by the World Health Organization as DHF/DSS[12]. This probability is known to depend on several factors, including host genetics[29], host immune status[14], and infecting virus genotype and serotype[30–32]. Here, we operationally define virulence as peak viral load because high viral load has been associated with high hemoconcentration[25,26], thrombocytopenia[25], and a higher pleural infusion index[23], all criteria used to define DHF[33]. Peak viral load has been shown to differ between primary and secondary dengue infections, most likely due to differences in the host's adaptive immune response between these infections[23,24]. Peak viral load has also been shown to be variable within both primary and secondary infections, due to viral factors[24,26] and other host factors[25,26]. Dengue virulence, as operationally defined by peak viral load, therefore arises from a combination of host factors and viral factors.

Because the probability that severe dengue disease results in death is low, estimated at less than one percent[12], disease-induced mortality is unlikely to play a role in limiting the evolution of virulence in dengue. We therefore first aimed to determine whether high viral loads during dengue infections are associated with shortened durations of infection. A negative association between viral load and the duration of infection could lead to a trade-off that would result in evolution towards intermediate virulence. Because dengue is an acute infection, viral load changes dramatically over time and the importance of the duration of infection is modulated by the degree to which an individual is infectious at different timepoints over this period. To determine whether a trade-off exists between dengue viral load and the duration of infection, we thus sought to examine the empirical relationship between peak viral load and the viral clearance rate. To this end, we calculated peak viral loads and viral clearance rates from viremia measurements of 239 symptomatic dengue-infected individuals (Methods). We found a positive relationship between individuals with detectable peak viral load and the viral clearance rate in both primary and secondary dengue infections

(Fig. 1). These findings are consistent with previous analyses showing that severe dengue infections are associated with both higher viral load peaks and higher viral clearance rates than non-severe dengue infections[23,24].

**Host immune response dynamics reproduce empirical trade-off.** To address the mechanism underlying this empirical trade-off, we fit an existing within-host model for dengue[34,35] to the longitudinal viremia measurements from all 239 dengue-infected individuals (Methods; Fig. 2a, b). The within-host model we used was chosen based on its ability to reproduce characteristic features of dengue infections and its consistency with findings from immunological studies in mouse models[36,37]. The statistical model fit incorporates inter-individual heterogeneity in the viral infectivity rate, which we included to capture variation in viral dynamics arising from host factors specifically, such as host immune status and host genetics (Methods). The statistically parameterized model reproduces the positive relationship between peak viremia and viral clearance rate observed in the data (Fig. 2c, d). It ascribes this positive relationship to the feedback between viral dynamics and the activation of the host immune response, with high viremia levels eliciting a strong innate and/or adaptive immune response, which in turn drives a high viral clearance rate.

It is well known that the overwhelming majority of dengue infections are asymptomatic infections[11]. Because the viral load data set consists exclusively of symptomatic dengue cases, which are known to have higher viral loads[38], a model fit to these data will overestimate peak viral load for average dengue infections. In order to consider asymptomatic infections, we re-fit the model to a subset of the data consisting of patients with lower viral loads (Methods). Simulations of this re-parameterized model reproduced viral load dynamics more consistent with both non-severe and asymptomatic dengue infections (Supplementary Fig. 1), while still reproducing the observed empirical trade-offs (Fig. 2c, d). The ability of these within-host models to quantitatively reproduce the trade-offs shown in Fig. 1 supports the hypothesis that the interaction between dengue virus and the host immune response is responsible for generating the empirical trade-offs observed in both primary and secondary dengue infections.

**Dengue virulence evolution.** To consider the potential for dengue virulence evolution, we next use our parameterized model to ask how selection may act on viral phenotypes that impact within-host dynamics. We specifically consider the viral production rate, which quantifies the rate at which viral progeny are produced from infected host cells, as an evolvable trait. We consider this viral phenotype because dengue strains are known to exhibit natural variation in this trait[39,40] and because this phenotype has in many cases been shown to have a viral genetic basis[39,41]. Further, this phenotype is known to impact the probability of developing severe disease, with viral strains with higher viral production rates being associated with higher virulence[13,40]. In our analyses below, we use the within-host model parameterization fit to the data subset, since this model parameterization we believe more appropriately reflects the dynamics of both asymptomatic and symptomatic infections, rather than only symptomatic infections.

We first consider the effects of the viral production rate on dengue virulence (as ascertained by peak viral load) and on transmission fitness in the context of primary dengue infections. Figure 3a shows how peak viral load depends on the viral production rate, as predicted by the within-host model. The predicted positive relationship between peak viral load and the viral production rate is consistent with empirical findings that higher viral production rates result in higher probabilities of triggering severe disease[39]. Using an empirically estimated relationship between host viral load and transmission probability to dengue's mosquito vector[22], we next predicted the relationship between the viral production rate and dengue's transmission potential (Methods), which is proportional to the basic reproduction number $R_0$. Figure 3b shows this relationship, with transmission potential being maximized at intermediate viral production rates, and therewith at intermediate peak viral loads. Figure 3c combines these results to more clearly demonstrate that dengue's transmission potential from individuals experiencing a primary infection is maximized at intermediate peak viral loads, reflecting intermediate virulence. Maximization of dengue's transmission potential at an intermediate viral production rate results from a trade-off between peak viral load and the viral clearance rate in primary infections (Supplementary Fig. 2), analogous to the trade-off that was reproduced by the within-host model incorporating inter-individual variation in dengue's viral infectivity rate (Fig. 2c). This within-host trade-off scales up to a transmission fitness trade-off because viral transmission success to dengue's mosquito vector saturates at high host viremia, while fitness costs associated with accelerated viral clearance continue to rise at higher viral production rates. Note that the results shown in Fig. 3 assume that the viral production rate only

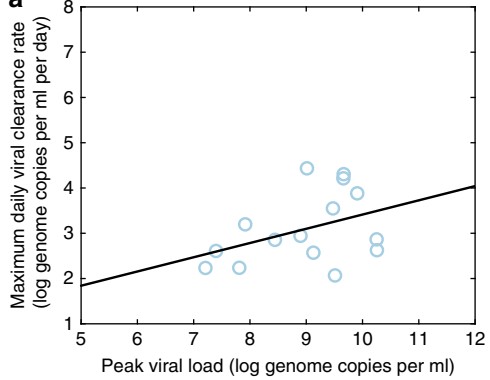
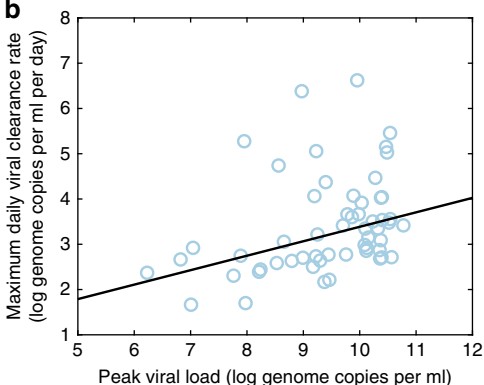

**Fig. 1** Empirical relationship between peak viral loads and viral clearance rates in dengue. **a** Peak viral loads plotted alongside maximum daily viral clearance rates for each of the individuals in the Ho Chi Minh City (HCMC) data set experiencing a primary dengue infection with a detectable viral peak ($n = 15$). The relationship is positive with a slope of $m = 0.33$, trending towards significance ($p = 0.14$, $t$-test). **b** Peak viral loads plotted alongside maximum daily viral clearance rates for each of the individuals in the HCMC data set experiencing a secondary dengue infection with a detectable viral peak ($n = 51$). The relationship is positive with a slope of $m = 0.32$ ($p < 0.01$, $t$-test)

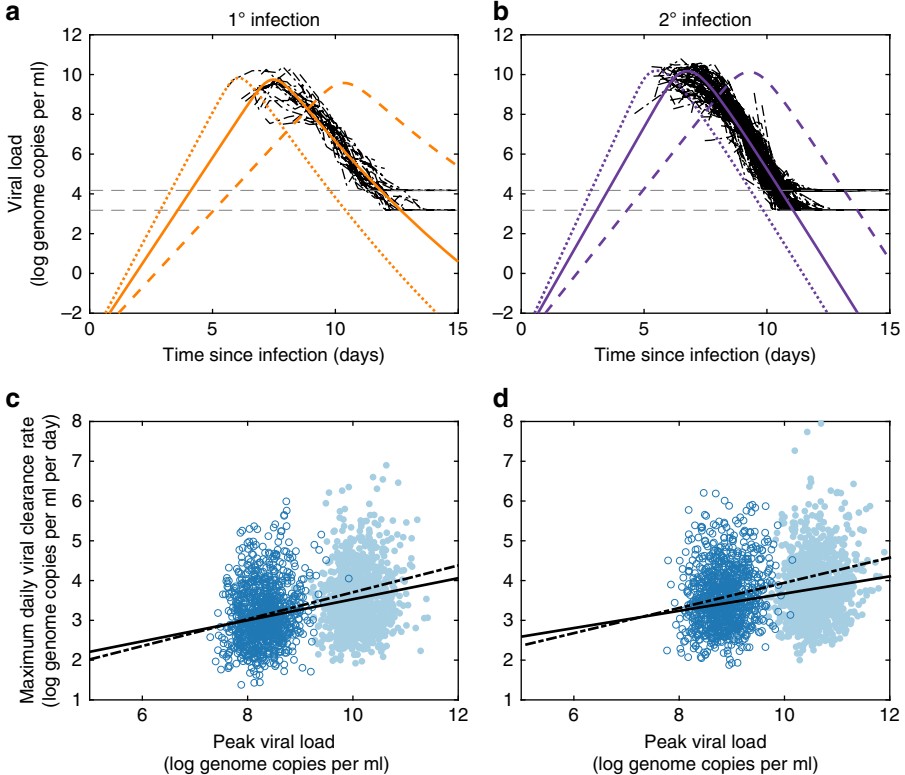

**Fig. 2** Within-host model dynamics reproduce empirically observed peak viral load-clearance rate trade-off. **a**, **b** Model-simulated viral load dynamics alongside viremia measurements from individuals experiencing **a** a primary dengue infection and **b** a secondary dengue infection. Solid colored lines show viral load dynamics parameterized with the mean estimated viral infectivity rate. Dotted lines show viral load dynamics parameterized with a viral infectivity rate two standard deviations above the estimated mean. Dashed lines show viral load dynamics parameterized with a viral infectivity rate two standard deviations below the estimated mean. Viremia measurements are shown in black and are shifted in time based on inferred individual incubation periods using model simulations with mean estimated viral infectivity rates. Viral limits of detection are shown with horizontal gray lines. **c** Model-simulated peak viral loads and corresponding maximum daily viral clearance rates in primary infections. Filled dots show results from the model parameterized using the full data set (simulated $n = 1000$). Open dots show results from the model parameterized using the data subset (simulated $n = 1000$). Both relationships are positive: slope $m = 0.26$ for the full data set (solid line, $p < 0.01$, $t$-test); slope $m = 0.34$ for the data subset (dashed line, $p < 0.01$, $t$-test). **d** Model-simulated peak viral loads and corresponding maximum daily viral clearance rates in secondary infections. Filled and open dots are as in **c**, each with simulated $n = 1000$. Both relationships are positive: slope $m = 0.21$ for full data set (solid line, $p < 0.01$, $t$-test); slope $m = 0.29$ for the data subset (dashed line, $p < 0.01$, $t$-test). Peak viral loads and viral clearance rates were calculated from model simulations incorporating inferred variation in the viral infectivity rate and following introduction of measurement noise. Model parameters are provided in Supplementary Table 1

impacts viral load dynamics in the human host, and therewith, the transmission success of dengue virus from humans to its mosquito vector. Our shown results implicitly assume that the viral production rate does not impact the probability that an infected mosquito transmits the virus to a susceptible host, an assumption we return to in the "Discussion" section.

Similar to our findings for primary infections, peak viral loads in secondary infections are also higher at higher viral production rates (Fig. 3a). Dengue's transmission potential is again maximized at an intermediate viral production rate (Fig. 3b) due to a trade-off between peak viral load and the viral clearance rate. However, the viral production rate that maximizes dengue's transmission potential is substantially lower in a secondary dengue infection than in a primary dengue infection. This indicates that secondary dengue infections select for viral production rates leading to lower peak viral loads in any given infection, relative to those selected for by primary dengue infections. This perhaps counterintuitive result stems from secondary infections generally having higher peak viral loads than primary infections[23,24]. As such, the transmission benefits of higher viral production rates are reduced for secondary infections relative to primary infections. The clearance rate costs of higher viral production rates are also higher in secondary infections than

in primary infections because the adaptive immune response plays an important role in clearing secondary dengue virus infections. If we instead assume, as for primary infections, that it is the innate immune response that clears secondary infections (Supplementary Fig. 3), transmission potential is still maximized at lower viral production rates than in primary infections (Supplementary Fig. 4), but in this case, transmission potential is maximized at viral production rates that yield a viral peak of approximately 7 log genome copies/ml (rather than 6 log genome copies/ml, as seen in Fig. 3c). Our finding that secondary dengue infections select for viral production rates associated with lower peak viral loads than primary dengue infections is also robust to alternative adaptive immune response formulations, such as one that has T-cell activation rates saturate at high antigen levels (see Supplementary Note 1 and Supplementary Fig. 5). This finding also holds for the model parameterization using the full data set (Supplementary Fig. 6).

Given our finding that dengue's fitness trade-offs depends on host immune status, we here synthesize our results to consider the role of epidemiological context in shaping dengue virulence evolution. We examine virulence evolution in two distinct epidemiological contexts: in a population with only a single dengue serotype endemically circulating and in a population with

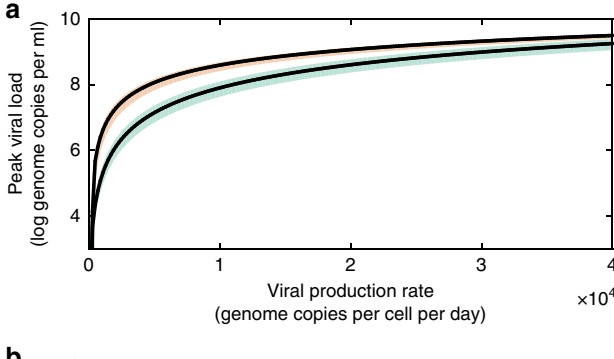

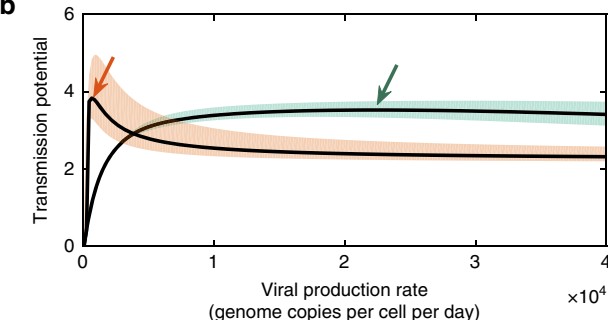

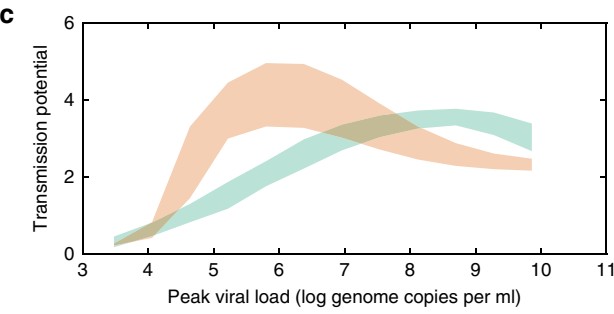

**Fig. 3** Model-predicted virulence trade-offs for dengue virus. **a** Model-predicted relationship between the dengue virus production rate and peak viral load, a strong predictor for the risk of developing severe dengue disease. **b** Model-predicted relationship between the dengue virus production rate and dengue's transmission potential. Green and orange arrows point to the peaks in transmission potential for primary and secondary infections, respectively. **c** Model-predicted relationship between peak viral load and dengue's transmission potential. In **a**–**c**, shaded regions show uncertainty estimates, derived from variation in the viral infectivity rate (Methods). Primary infections are shown in green and secondary infections are shown in orange

two serotypes circulating. To determine the viral production rate at which viral fitness is maximized in each of these epidemiological contexts, we use an adaptive dynamics framework[42] (Methods).

When only one serotype is circulating, all infections are primary infections, and the evolutionary stable viral production rate is one that maximizes the transmission potential of a primary dengue infection (Fig. 4a). This optimal viral production rate is not impacted by the transmission intensity of the virus (Fig. 4b), which may differ because of geographic differences in mosquito densities or in the mosquito biting rate. This is because, in a situation with only one serotype circulating, all infections are primary infections, regardless of the extent of dengue endemism. In a population with two circulating dengue serotypes, the evolutionary stable viral production rate is always lower than in the context of only one circulating serotype (Fig. 4b). This is because secondary infections select for lower viral production

rates (Fig. 3b). As such, at evolutionary equilibrium, primary infections are expected to result in a lower viral peak (and therefore a lower probability of triggering severe dengue disease) in a population with two circulating serotypes than in a population with only one serotype circulating (Fig. 4c). In a population with two circulating serotypes, however, the transmission intensity of the virus does impact the optimal viral production rate (Fig. 4b), since this transmission intensity impacts the proportion of dengue infections that are primary versus secondary infections. At higher transmission intensities, and under the assumption of no heterologous cross-immunity between the serotypes, the proportion of dengue infections that are secondary infections relative to primary infections increases, saturating at 50%. Under this assumption, the optimal viral production rate therefore decreases with higher transmission intensities. Optimal viral production rates are to some extent impacted when a period of heterologous cross-immunity is assumed between the circulating serotypes (Fig. 4b). When cross-immunity acts to temporarily shield an individual from infection, lower viral production rates are selected for at higher transmission intensities, similar to the case with no heterologous cross-protection. However, under the assumption that cross-immunity shields not from infection, but only from symptomatic disease ("clinical cross-protection"[43]), optimal viral production rates change non-monotonically with transmission intensity (Fig. 4b). In all cases, however, populations with two circulating serotypes are expected to select for lower viral production rates than populations with only a single circulating serotype. The within-host model parameterized using the full data set yields qualitatively similar results (results not shown).

## Discussion

Here we have shown empirical support for a trade-off between peak viral load and the viral clearance rate in dengue, the most prevalent vector-borne viral disease of humans. Using a within-host model fit to viremia measurements from symptomatic dengue-infected individuals, we found that this trade-off can be explained by the interaction of the virus with the host immune response, providing empirical support for this hypothesis from theoretical studies[10,21]. Our findings are further consistent with an analysis of the within-host dynamics of arboviruses (including dengue) in experimentally challenged vertebrate hosts[44]. That analysis showed that arboviruses appear to exhibit a trade-off between peak viremia and the duration of infection, a trade-off that was made apparent when considering various doses of virus administered[44]. This study further invoked immune clearance as the likely mechanism for the observed trade-off, given that target cell limitation is unlikely, especially for dengue.

We next considered whether a transmission-clearance trade-off analogous to the trade-off we observed empirically could result in selection pressures on dengue virus to evolve to intermediate virulence. We specifically considered the possibility for dengue virus to evolve its viral production rate. By combining within-host model simulations with empirical estimates of transmission success to dengue's mosquito vector, we found that viral fitness, as measured by transmission potential, is maximized at intermediate dengue virus production rates. These intermediate production rates are associated with intermediate peak viral loads and thus an intermediate risk for triggering severe disease (i.e., intermediate virulence). We further found that the viral production rate at which transmission potential was maximized differed by immune status, with secondary infections selecting for lower viral production rates than primary infections. Because lower viral production rates result in lower peak viral loads, this means that secondary infections select for genetically less virulent viruses.

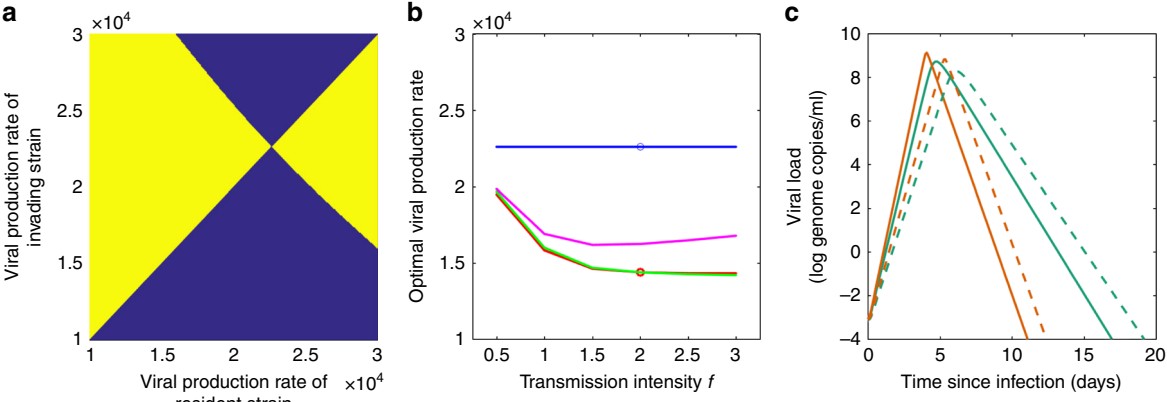

**Fig. 4** Evolutionary stable viral production rates depend on epidemiological context. **a** Pairwise invasibility plot for a population with a single endemically circulating serotype. The *x* axis plots viral production rates of the resident dengue strain; the *y* axis plots viral production rates of the invading strain. Yellow (blue) regions show viral production rates that can (cannot) successfully invade populations with the resident strain present. The evolutionarily stable strategy occurs at the viral production rate where neither a higher production rate phenotype nor a lower viral production rate phenotype can invade. **b** Evolutionary stable viral production rates as a function of dengue transmission intensity *f*. Differences in transmission intensities reflect geographical differences in mosquito densities or biting rates. Optimal viral production rates are shown under two epidemiological contexts: when only a single serotype is circulating (blue line) and when two serotypes are circulating. For the two serotype scenarios, we show optimal viral production rates in the absence of heterologous cross-protection (red line), when the period of "classical" cross-protection is assumed to be 2 years long (green line), and when the period of "clinical" cross-protection is assumed to be 2 years long (magenta line). At any transmission intensity setting, the evolutionarily stable viral production rate is lower in a population with two serotypes circulating rather than only one. The open circles denote the evolutionary stable viral production rates used in subplot **c**. **c** Within-host dengue virus dynamics, simulated with two evolutionary stable viral production rates. Green lines show primary infection simulations; orange lines show secondary infection simulations. Solid lines show simulations with the optimal viral production rate in the population with one endemically circulating serotype ($\omega = 22{,}600$ genome copies per cell per day). Dashed lines show simulations with the optimal viral production rate in the population with two endemically circulating serotypes and no heterologous cross-immunity ($\omega = 14{,}380$ genome copies per cell per day). For the sake of completeness, we show secondary infection viral dynamics with the viral production rate that is evolutionary stable in the population with one circulating serotypes, although this epidemiological context does not have secondary infections

Finally, as a consequence of these immune-status effects, we have shown that epidemiological context impacts the viral production rate that is evolutionary stable, with single-serotype epidemiological settings selecting for higher within-host viral production rates than epidemiological settings with two circulating serotypes. Our conclusions are robust to two different within-host model parameterizations, one fit to the full data set and one fit to a data subset of lower viral loads that attempts to capture asymptomatic infections.

Our finding that dengue virus should evolve towards intermediate virulence depends on the formulation of our within-host model, specifically on our assumptions of how the host's immune system responds to viral infection. The within-host model we fit is based on an existing within-host model for dengue virus that has been shown to successfully reproduce characteristic features of dengue infections, including the timing and magnitude of peak viremia, and the viral clearance rate, as well as immunological features of dengue infections[34]. Furthermore, previous work has shown that this model can recover findings from virological and immunological studies of dengue infection when statistically fit to viral load data[35]. As such, although the fitness trade-offs derived here depend on a within-host model formulation, the model formulation we have used is one that has considerable empirical support. With the use of this model, a transmission fitness trade-off occurs because both the innate and adaptive immune responses are activated in an antigen-dependent manner. High levels of viremia therefore result in strong activation of the immune response and a shorter duration of infection (consistent with the patterns shown in Fig. 1). A model in which the immune response is generated in an antigen-independent manner would not generate a transmission-virulence trade-off because no costs would be incurred by a virus for an increase in its viral production rate. However, a completely antigen-independent stimulation

of the immune response is unlikely, given the observed within-host trade-offs (Fig. 1), as well as existing empirical findings that show a relationship between the strength of innate and adaptive immune responses and viral load and/or the severity of dengue disease[25,45–47].

Our finding of a transmission fitness trade-off in dengue also critically depends on quantitative estimates of how within-host viral load impacts the probability that a susceptible mosquito becomes infected following a bite from an infected host[22]. While our model has incorporated this demonstrated relationship between host viral load and transmission success to the mosquito, our model implicitly made several other simplifying assumptions. First, we assumed that host viral load only impacted transmission success to the mosquito, but not viral dynamics within the mosquito, which may in turn affect transmission success from mosquito to host. When we considered selection on the dengue virus production rate, we further assumed that this viral trait only affected viral dynamics in human hosts. It could be the case that the viral production rate also impacts viral dynamics in the mosquito vector, and therewith potentially onward transmission success to susceptible hosts. Indeed, several studies have indicated that more virulent dengue virus genotypes may have a fitness advantage in mosquitoes. Specifically, Hanley and colleagues showed that a more virulent genotype of DENV-3 reached higher titers in the midgut of the primary mosquito vector of dengue virus, *Aedes aegypti*, relative to a less virulent DENV-3 genotype[48]. They further showed that the more virulent genotype more effectively disseminated throughout the body of the mosquito vector, indicating that the probability that an infected mosquito successfully transmits the virus to a susceptible human host is likely higher for the more virulent genotype compared to the less virulent genotype. Cologna and colleagues also found that more virulent dengue genotypes replicated more efficiently in

mosquito vectors[40]. Finally, OhAinle and colleagues found that a dengue clade associated with higher viremia in a hospital study had a fitness advantage during viral competition assays in a mosquito cell line[49]. The findings of these studies can be incorporated into our virulence evolution analysis by assuming that higher viral production rates increase the probability of dengue transmission success from infected mosquito to susceptible host (Supplementary Note 3). Incorporation of this fitness benefit in mosquitoes would lead to higher optimal viral production rates in both primary and secondary infections (Supplementary Fig. 8). Finally, our model did not consider the effect that within-host viral dynamics, or host symptom manifestation, may have on the contact rate between hosts and vectors. Incorporating these "coupled heterogeneities"[50], once quantified, into virulence evolution analysis is critical for accurately assessing the evolutionary pressures on viral traits.

Theory on virulence evolution when pathogens cause disease via immunopathology has focused on how virulence evolution depends on the relationship between immunopathology and parasite exploitation[51], and the effect of immunopathology on epidemiological traits[52]. Our analysis adds another piece of complexity to these theoretical studies by showing that exposure history can impact virulence evolution in a case such as dengue, where the probability of developing severe disease through immunopathology depends strongly on immune status. Specifically, we have shown that we expect virulence to evolve to lower levels when secondary infections are more common. Conversely, when primary infections predominate, more virulent viral phenotypes have a fitness advantage. We here only inferred fitness trade-offs in primary and secondary dengue infections. This is because within-host dynamics of post-secondary dengue infections have not been well-characterized, as very few symptomatic post-secondary infections typically present in hospitals[17]. The few existing studies on post-secondary infections[53–55] indicate that the reduced risk of triggering severe disease during a post-secondary infection is associated with a protective immune response[53,54], and a study in non-human primates showed that tertiary dengue infections resulted in lower viremia levels than primary and secondary infections[55]. Taken together with studies that have shown that high viral load is associated with severe disease[23,24,26], these studies indicate that peak viral load in post-secondary infections is likely lower than in primary and secondary infections. This suggests that post-secondary infections may select for highly virulent dengue strains as the benefit of higher viral peaks in these infections will exceed the benefits of higher viral peaks in primary and secondary infections. The cost of increasing viremia in post-secondary infections, however, is unknown and will necessarily depend on the strength of the innate and adaptive immune responses in post-secondary infections and how these responses affect the duration of infection.

Beyond the need to characterize the role of post-secondary infections in dengue virus virulence evolution, more work needs to be done to understand the selection pressures that asymptomatic cases place on this virus. A recent study characterizing whether mosquitoes could become infected by biting a host with an asymptomatic dengue infection showed that, controlling for viral load, asymptomatic individuals were more infectious to mosquitoes than were symptomatic individuals[38]. Although this study included only 13 asymptomatic individuals, this work has important implications for the role of asymptomatic infections in driving dengue transmission patterns and dengue virulence evolution if the proportion of asymptomatic infections differs between primary, secondary and post-secondary infections. In a cohort study in Nicaragua, no difference was found in the proportion of inapparent versus symptomatic infection by immune status[56], suggesting that including differences in asymptomatic and symptomatic transmissibility into our analysis may not impact our conclusions.

Heterogeneity in individual response to dengue infection is more complicated than simply whether an individual is asymptomatically or symptomatically infected with dengue. While we have here focused on the role that immune status plays in viral dynamics and, relatedly, the probability of developing severe disease, we also considered interindividual variation in viral load dynamics for individuals with the same immune status. Specifically, our model incorporates a random effect in the viral infectivity rate, which can differ between individuals because of host genetics or other reasons. Consistent with theory[57], incorporating this within-host heterogeneity does not affect our general conclusion that viral transmission fitness is maximized at intermediate virulence.

Our analyses show that, for dengue, understanding virulence evolution is incomplete without explicitly considering an epidemiological context. Specifically, we found that regions with only a single serotype present will select for a more virulent within-host phenotype than regions with two serotypes circulating. These findings may be consistent with the emergence of virulent DENV-2 genotypes in many countries in Latin America in the late 20th century[58] where one serotype typically predominated at a time. Further characterization of post-secondary infections are needed to consider hyperendemic scenarios with four circulating serotypes. A hyperendemic scenario may select for higher or lower virulence relative to a two-circulating serotype scenario depending on whether optimal viral production rates of post-secondary infections are more similar to primary or secondary infections. The epidemiological contexts on which we have focused are also all endemic contexts, where viruses maximize fitness by maximizing their basic reproduction numbers. However, transient dengue epidemics may select for within-host viral phenotypes that instead maximize the intrinsic population growth rate of the virus[59–61]. This suggests that viral virulence may evolve to higher levels in both primary and secondary infections than predicted by our endemic analyses. We would still expect, however, that primary dengue infections select for more virulent phenotypes than secondary dengue infections in these epidemic contexts.

Our results indicate that geographic regions will likely differ in the evolutionary selection pressures they place on dengue virus, and future work should focus on identifying the regions that will most likely be the sources of more virulent dengue strains. Empirically characterizing a region's epidemiological context is complicated by the difficulty in detecting asymptomatic infections and distinguishing between secondary and post-secondary infections. However, cohort studies that detect inapparent infections such as in Vietnam[38] and the long-running longitudinal pediatric dengue cohort study in Nicaragua[62] may be useful in characterizing epidemiological context more accurately. Latin America is an especially good area for studying the links between viral load kinetics and onward transmission potential because in many Latin American countries, only one serotype predominates over time, possibly decreasing the complexity of epidemiological context.

In the 21st century we have seen a shift to hyperendemism in many countries around the world, as well as the emergence of dengue in new regions[29]. While many epidemiological studies focus on case severity, our results here show that regions with low case severity may play a critical role in incubating highly virulent dengue strains. As the global distribution of dengue shifts, understanding the changing patterns of infection histories is crucial for predicting sources of virulent dengue strains. More broadly, these results provide data-driven support for how

complex patterns of immunity can have large effects in shaping pathogen virulence evolution.

## Methods

**Data**. The empirical data set of viremia measurements from 239 symptomatic dengue-infected individuals originates from a clinical trial of adult dengue patients at the Hospital for Tropical Diseases in Ho Chi Minh City (HCMC), Vietnam[24,63]. The trial studied the effects of the antiviral drug chloroquine on dengue, but the drug had no measurable effect on viral load dynamics[63]. As in previous analyses of this data set[24,35,64,65], we therefore do not distinguish between chloroquine-treated patients and control patients. Patients were admitted into the study if their first viral load measurement was within 72 h of the onset of symptoms. Viral load was then measured twice per day using RT-PCR, with the assays having limits of detection of either 1500 genome copies/ml or 15,000 genome copies/ml[64]. The data set consists of individuals infected with one of dengue's four serotypes, where infected individuals were further stratified by immune status (primary infection or secondary heterologous infection), and clinical manifestation (DF or DHF), as described previously[24]. Note that classification of immune status did not include post-secondary infection so some individuals classified with a secondary infection may in fact have been experiencing a post-secondary infection.

For each individual, the maximum daily viral clearance rate was calculated by doubling the maximum decrease in viral load (on the $\log_{10}$ scale) between two consecutive viremia measurements (taken 12 h apart). We then calculated the maximum viral load for each individual. Because the onset of dengue symptoms often occurs at or following peak viral load, many of the dengue-infected individuals had viremia measured only during viral decline, with no detectable viral peak. We categorized dengue-infected individuals as either having or not having a detectable viral peak, based on whether viral load was observed to increase following the first viremia measurement. We fit the relationships between maximum viral load and viral clearance rate shown in Fig. 1 using only the subset of individuals who had detectable viral peaks. Note that some of the individuals in this subset may have inappropriately been categorized as having a detectable viral peak due to measurement error or to intrinsic fluctuations in viral dynamics. Following an initial regression, we computed Cook's distance to identify outliers in this data set. Datapoints from two individuals in the primary dengue infection data set and datapoints from five individuals in the secondary dengue infection data set were above the threshold value of three times the mean Cook's distance so we refit the relationships between maximum viral load and the viral clearance rate once these datapoints were excluded.

We further examined the correlation between peak viral load and the viral clearance rate slope in the subset of individuals having detected viral peaks. In this case, we fit the viral clearance rate slopes using the likelihood function described previously in ref. [34]. This alternative analysis again yielded a positive relationship between peak viral load and viral clearance rate in secondary dengue infections, with slope $m = 0.12$ ($p < 0.01$, t-test). This approach, however, did not yield a significantly positive relationship between peak viral load and viral clearance rate in primary infections.

**Within-host model**. The within-host model we parameterize using the viremia measurements is given by:

$$
\begin{aligned}
\mathrm{d}X/\mathrm{d}t &= -\beta X V \\
\mathrm{d}Y/\mathrm{d}t &= \beta X V - \alpha N Y - \delta_T Y T \\
\mathrm{d}V/\mathrm{d}t &= \omega Y - \kappa V \qquad\qquad (1) \\
\mathrm{d}N/\mathrm{d}t &= q Y - d_N N \\
\mathrm{d}T/\mathrm{d}t &= q_T Y T
\end{aligned}
$$

where the variables are uninfected target cells $X$, infected target cells $Y$, free virus $V$, natural killer (NK) cells $N$, and T cells $T$. NK cells capture the role of the innate immune response. T cells capture the role of the cellular immune response. In the model, uninfected cells become infected with free virus at an overall rate of $\beta X V$. Infected cells are cleared by NK cells at an overall rate of $\alpha N Y$ and by T-cells at an overall rate of $\delta_T Y T$. Virus is produced by infected cells at an overall rate of $\omega Y$ and cleared at rate $\kappa V$. NK cells are activated at an overall rate $q Y$ and decay at rate $d_N N$. T cells are activated at an overall rate of $q_T Y T$. The parameter $\beta$ is known as the viral infectivity rate. The parameter $\omega$ is known as the viral production rate. Supplementary Table 1 provides the list of model parameters and variables, as well as short descriptions of each.

The model is based on an existing within-host model for dengue virus that has been shown to successfully reproduce characteristic features of dengue infections, including the timing and magnitude of peak viremia and the viral clearance rate[34]. Here, we do not include a T-cell death rate because fitting the within-host model under the assumption of a positive T-cell death rate does not appreciably change inferred model parameter values and other within-host models for acute infectious diseases often ignore this T-cell death rate (e.g., refs. [64,66,67]). By setting the initial number of T cells to 0 in primary infections, we do not explicitly include a cellular immune response during these infections, consistent with the virological finding that primary dengue infections do not require a cellular immune response to be cleared[36]. Although not explicitly modeled, we assume that during a primary

infection a cellular immune response kicks in at the end of infection such that viremia cannot increase again once it has fallen below the limit of detection.

**Within-host model parameterization using full HCMC data set**. Previously, we used a Markov chain Monte Carlo (MCMC) approach to fit Eq. (1) to the viral load data set described above for individuals symptomatically infected with DENV-1, DENV-2, and DENV-3[35]. We estimated parameters for models of varying complexity based on hypotheses reflecting virological and immunological studies of dengue infection and compared model fit using Bayesian model selection criteria[35]. The models that best fit the data were models in which the viral infectivity rate $\beta$ differed by clinical manifestation (DF vs. DHF) and by infecting serotype. These previous results suggest that a significant portion of the variation in dengue viral load can be explained by differences in the viral infectivity rate, reflecting differences in host factors such as host genetics and immune status, and to some extent differences in infecting serotype.

To accommodate interindividual variation in viral infectivity rates that originates from host factors unrelated to immune status, we here include a random effect on the viral infectivity rate $\beta$. To further allow for the possibility that viral infectivity rates differ systematically by immune status (primary or secondary dengue infection), we let viral infectivity rates during both primary infections and secondary infections be normally distributed with different means $\beta_{PI}$ and $\beta_{SI}$, but with the same coefficient of variation. Differences in viral infectivity rates by immune status reflect the process of ADE. We therefore expect that the mean viral infectivity rate in secondary infections ($\beta_{SI}$) will be higher than that in primary infections ($\beta_{PI}$).

We further allow the innate immune response activation rate $q$ to differ between primary and secondary infections based on evidence that antibody–virus complexes formed during secondary infections lead to suppression of the innate immune response ("intrinsic ADE")[68–71]. We therefore expect the innate immune response activation rate in primary dengue infections $q_{PI}$ to be higher than in secondary infections $q_{SI}$.

We fit the model simultaneously to the viremia measurements from primary and secondary infections. Due to parameter unidentifiability described in ref. [35], we assign values to certain parameters and estimate the values of others (Supplementary Table 1). Because we include interindividual variation in $\beta$, the viral clearance rate $\kappa$ and the initial level of free virus $V_0$ also become practically unidentifiable. We therefore also set $\kappa$ and $V_0$ based on our previous estimates in ref. [35] (Supplementary Table 1).

As in our previous analysis, we include interindividual variation in the incubation period IP, defined as the time between viral inoculation and the onset of symptoms by including a log-normally distributed random effect on IP with mean $\log(\mathrm{IP}_g)$ and standard deviation $\sigma_I$. We set both $\mathrm{IP}_g$ and $\sigma_I$ based on existing empirical estimates[72] (Supplementary Table 1).

We initially fit seven parameters using the viremia measurements from the 239 infected dengue cases (a total of 2485 viremia measurements): $\beta_{PI}$, $\beta_{SI}$, $q_{PI}$, $q_{SI}$, the rate of NK cell decay $d_N$, the rate of T-cell activation $q_T$, and the coefficient of variation in the viral infectivity rate $c$. Our likelihood expression, based on the one presented in[35], assumes that viremia measurement error is normally distributed (with standard deviation $\sigma_\epsilon$; Supplementary Table 1) on the $\log_{10}$ scale and accommodates measurements below the limit of detection. Our likelihood expression is a basic extension of the one presented in ref. [35], which numerically integrates out not only interindividual variation in incubation periods, but also interindividual variation in viral infectivity rates. Because we do not have any prior information on any of the parameters we seek to estimate and because evaluating the likelihood expression is computationally very intensive, we use maximum likelihood estimation instead of MCMC to fit the model to the viremia data set. Specifically, we use the fminsearch optimization function in MATLAB to minimize the negative log-likelihood.

Due to lack of immune data and early infection viral load data, we were unable to robustly estimate the relative contributions of the innate and adaptive immune response during secondary infections. The model likelihood is maximized at a low $q_{SI}$ value, consistent with our expectation that $q_{SI} < q_{PI}$. We used a likelihood ratio test to determine whether a simpler model with $q_{SI} = 0$ can be rejected, and we failed to reject this hypothesis. This indicates that given the current data, the T-cell response alone can sufficiently reproduce the faster viral clearance rate observed in secondary infections relative to primary infections. Therefore, the secondary infection model reduces to a model with only the cellular adaptive immune response responsible for clearing infection and we do not include NK cells in the secondary infection model. In total, we therefore fit only six parameters to the entire data set: $\beta_{PI}$, $\beta_{SI}$, $q_{PI}$, $d_N$, $q_T$, and $c$, and set $q_{SI}$ to 0. We expect that additional data would show that the innate immune response in secondary infections plays a contributing role for regulating dengue virus dynamics, especially early in infection. Note that we do not estimate six parameters for each individual. Rather, we estimate a total of six parameters using the entirety of the data set.

Maximum likelihood estimates of the six fitted parameters (with 95% confidence intervals) are provided in Supplementary Table 1. To determine confidence intervals we computed slice likelihoods for each parameter by varying one parameter while all other parameter values were set at their maximum likelihood values[73]. The 95% confidence interval is equivalent to a decrease of 2 in log-likelihood from the maximum log-likelihood[74].

  

As expected from the effects of ADE[14,75], $\beta_{SI}$ was estimated to be higher than $\beta_{PI}$ (27% higher). A higher viral infectivity rate and a lower innate immune response activation rate ($q_{SI} = 0$) results in secondary dengue infections having higher peak viral loads than primary dengue infections (on average, peak viral loads of 10.2 log genome copies/ml compared to 9.7 log genome copies/ml, in the absence of measurement noise). Given our operational definition of virulence, these results indicate that secondary infections should be at higher risk for triggering severe disease than primary infections, consistent with epidemiological studies[23,24,26]. Further, viral clearance rates in secondary infections exceed those in primary infections (1.9 log genome copies/ml/day compared to 1.4 log genome copies/ml/day, in the absence of measurement noise), consistent with virological studies[23,24].

**Within-host model parameterization using HCMC data subset.** Longitudinal cohort studies indicate that only 6–40% of infections are symptomatic, with the remainder being inapparent[56,76]. Viral load measurements from asymptomatic infections are generally not available, although a recent study has shown that asymptomatic infections exhibit lower viremia than symptomatic infections[38]. In an attempt to consider viral load measurements representative of both asymptomatic and symptomatic infections, we re-fit the within-host model given by Eq. (1) to a subset of the HCMC data set. This subset consists of individuals with lower viral loads, specifically, individuals experiencing a primary infection whose maximum viral load was ≤8 log genome copies/ml and individuals experiencing a secondary infection whose maximum viral load was ≤8.5 log genome copies/ml. These viral load levels correspond to the middle range of peak viremia values typically seen in symptomatic dengue infections[23–26,77]. A difference of 0.5 log genome copies/ml between peak viral loads of primary versus secondary infections was chosen for consistency with findings in ref. [24]. This subset of the viral load data set includes 11 primary infections (37% of all primary infections) and 90 secondary infections (43% of all secondary infections), comprising a total of 1020 viremia measurements.

We expect that the lower viral loads observed in this subset of individuals are due to lower viral infectivity rates $\beta$, and/or a stronger immune response (higher $q_{PI}$ and/or lower $d_N$ in primary infections, and higher $q_T$ in secondary infections) in these individuals compared to individuals in the full data set. Due to the small number of individuals in the data subset, and because we have no reason to expect that the coefficient of variation $c$, or the ratio of $\beta_{SI}/\beta_{PI}$, to differ between the full data set and the data subset, we fit only four parameters to the data subset: $\beta_{PI}$, $q_{PI}$, $q_T$, and $d_N$. We estimated a small value of $d_N = 1 \times 10^{-9}$/day, and therefore performed a likelihood ratio test to determine whether a simpler model with $d_N = 0$ could be rejected. We were unable to reject this simpler model and therefore set $d_N$ to 0. The final parameter estimates are provided in Supplementary Table 1.

**Quantifying population-level viral transmission potential.** In endemic situations, viral fitness is given by the basic reproduction number $R_0$, which for a vector-borne disease depends on the vector biting rate $b$, the ratio $m$ of female mosquitoes to humans, the lifespan of mosquitoes $1/\mu$, and the probabilities of an infected human (mosquito) transmitting the virus to a susceptible mosquito (human). The most general formulation of $R_0$ for a vector-borne disease is given in Equation (1) of ref. [78]. In this work, the authors assumed that the probability of an infected mosquito transmitting dengue to a susceptible human ($p_m(V|t)$) is proportional to $q_m(V|t)$, the proportion of mosquitoes that have detectable virus in their saliva $t$ units of time after having taken a blood meal from an infected human with viral load $V$. In turn, these authors did not find that $q_m(V|t)$ depended on viral load $V$. As such, their term, $\int_0^\infty p_m(V|t)e^{-\mu t}dt$ evaluates to a constant, which we call $p_{hm}$. This leads to the following expression for $R_0$:

$$R_0 = mb^2 p_{hm} \int_0^\infty \left( \int_0^\infty p_h(V|\tau)d\tau \right) p(V)dV \quad (2)$$

where $p(V)$ is the probability that an infected human with viral load $V$ transmits the virus to a susceptible mosquito upon being bitten and $p_h(V|\tau)$ is the probability that a human who was infected with dengue $\tau$ units of time ago has viral load $V$. Since we simulate our within-host model deterministically under any given parameterization, we can equivalently write this equation as:

$$R_0 = mb^2 p_{hm} \int_0^\infty p(V(\tau))d\tau \quad (3)$$

where $V(\tau)$ is the viral load of an infected human who was infected $\tau$ units of time ago. We call the integral inside this expression the transmission potential of the strain.

To calculate the transmission potential, we rely on a recent experimental study analyzing factors that influence transmission success of dengue virus to susceptible mosquitoes[22]. In this study, 407 susceptible mosquitoes were fed on 208 hospitalized dengue patients. An analysis of whether these blood-fed mosquitoes became infected with dengue (as measured by presence of the virus in their abdomens) showed that host viremia was the strongest covariate explaining successful human-to-mosquito infection[22]. Based on this finding, the authors quantified the relationship between patient viremia and the probability of human-to-mosquito transmission using marginal logistic regression for each of dengue's

four serotypes. We used this determined relationship to couple the viral load dynamics derived from our within-host model ($V(\tau)$) to the probability of viral transmission from infected human to susceptible mosquito $p(V(\tau)) = \frac{1}{1+e^{-(c_0+c_1 V(\tau))}}$. We refer the reader to ref. [22] for the experimental data and the parameters of the logistic regression curves. We used the DENV-1 parameterization from[22] because the majority of dengue infections in the HCMC data set are dengue serotype 1. Our results are robust to transmission probabilities derived for the other three dengue serotypes (Supplementary Note 2, Supplementary Fig. 7).

**Peak viral load and transmission potential.** In Fig. 3a, we plot the relationship between peak viral load and the viral production rate $\omega$. In Fig. 3b, we plot the relationship between transmission potential and $\omega$. The black solid lines in Fig. 3a, b show the peak viral loads and transmission potentials evaluated at mean viral infectivity rates, which depend on whether the infection is a primary or secondary infection. At each value of $\omega$ considered, we further sample 100 viral infectivity rates from the distribution with mean $\beta$ and standard deviation $c\beta$, where $c$ is the coefficient of variation estimate. For each of these sampled viral infectivity rates, we simulate the within-host model, calculate the peak viral load, and evaluate the transmission potential. In practice, based on the results in[22], we assume that when viral load levels are below 1500 genome copies/ml, transmission to the mosquito will not occur. The confidence intervals in Fig. 3a, b show the 95% range of values obtained from these 100 samples at each $\omega$ value. In Fig. 3c, we use the combined set of simulations across $\omega$ values. Specifically, using a sliding window of peak viral load levels, we find the simulations having peak viral loads in a given window and plot the 95% range in transmission potential values across this set of simulations.

**Virulence evolution in different epidemiological contexts.** We use an adaptive dynamics framework to consider dengue evolution of virulence in different epidemiological contexts. We specifically consider two different endemic contexts: one with a single circulating serotype and one with two circulating serotypes. The transmission dynamics under each of these endemic contexts are based on an existing epidemiological model for dengue. Specifically, the models we use are deterministic, age-structured epidemiological models with age-dependent mortality rates, as described in ref. [79] (the "Duke model"). All of the models we consider assume life-long immunity to reinfection with a homologous dengue serotype. The models differ in the type and duration of transient cross-protection that is assumed between heterologous serotypes. We consider three models: one with no heterologous cross-protection, one with a 2 year period of "classical" cross-protection (where cross-protection temporarily prevents infection with the heterologous serotype), and one with a 2 year period of "clinical" cross-protection (where cross-protection prevents symptom development and onward transmission, but does not prevent infection or, critically, seroconversion)[43]. We assume a 2 year period of cross-protection in the latter two models based on previous findings[16]. We do not include seasonality in the model.

For the single circulating serotype endemic context, the epidemiological model collapses to a basic age-structured susceptible-infected-recovered (SIR) model. In this case, the viral production rate that maximizes the transmission potential of primary infections is the evolutionary stable viral phenotype. To confirm this, we simulated this model with a "resident" dengue strain having an $R_0$ that is calculated from its viral production rate. Specifically, we calculate a strain's $R_0$ by multiplying its calculated transmission potential (which depends on $\omega$, Eq. (3)) by a transmission intensity factor of $f = m(b^2)p_{hm}$. Once simulations of this model have reached their epidemiological equilibrium, we fix the number of susceptible and recovered hosts at their equilibrium values and determine numerically whether an "invading" strain of the same serotype (which has its own viral production rate, and corresponding $R_0$) can increase when rare. If the invading strain can increase when rare, its reproductive rate $R$ exceeds 1 in the endemic context of the resident strain and (given full cross-immunity) it would replace the resident strain. The pairwise invasibility plot shown in Fig. 4a shows the results of this one-serotype analysis for combinations of "resident" and "invading" strains when $f$ is set to 2. This pairwise invasibility plot, however, does not change with a change in $f$ (Fig. 4b), given that all infections (regardless of their number) are primary infections.

For the two-circulating serotypes endemic context, the epidemiological models remain more complicated, but we adopt a conceptually similar approach to the one we used for the single-serotype endemic context to determine the evolutionary stable viral production rate. Specifically, for a given transmission intensity $f$, we generate a pairwise invasibility plot under the assumption that both "resident" strains have the same viral production rate, and therefore also the same $R_0$. We then simulate the model to its epidemiological equilibrium. We then fix the number of uninfected hosts (regardless of their history of infection) at their equilibrium values and determine numerically whether "invading" strains (which have their own viral production rate that is the same across the two circulating serotypes) can increase when rare. Figure 4b shows evolutionary stable viral production rates for these two-serotype epidemiological contexts across a range of $f$ values, reflecting different transmission intensities that are due to differences in mosquito densities ($m$) or biting rates ($b$).

**Code availability.** All code is available on GitHub at https://github.com/rbenshachar/denv_virulence_evolution.

**Data availability**. The viral load data were downloaded from ref. [64].

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

## Acknowledgements
This work was supported by the MIDAS CIDID Center of Excellence (U54-GM111274).

## Author contributions
R.B.-S. and K.K. jointly conceived the study. R.B.-S. performed within-host model and transmission potential analyses while K.K. performed the pairwise invasibility analyses. Both R.B.-S. and K.K. contributed to analyses and wrote the manuscript.

## Additional information

**Competing interests:** The authors declare no competing interests.

