## [Peer Review File · Nature Communications]

Reviewers' comments:

Reviewer #1 (Remarks to the Author):

In this study, Ben-Shachar and Koelle provide strong evidence for the existence of a key fitness trade-off that promotes the evolution of intermediate virulence in human dengue. Given the paucity of studies documenting such fitness trade-offs, relative to their importance in mathematical theory for pathogen evolution, this is a very important and novel study. Moreover, the study is comprehensive; I particularly appreciated the analysis of pathogen evolution across different epidemiological contexts (primary vs. secondary infection). The manuscript is very well-written, and the Discussion section touched on all of the major points that I wanted to see. In my opinion, this is an important study that will appeal to a broad audience, and thus warrants publication in Nature Communications.

My main comment is one that I am sure the authors considered in drafting the manuscript: why structure the Introduction and Discussion around transmission-virulence trade-off theory, when the data (and title) clearly supports a transmission-recovery trade-off? Both trade-offs have been shown to lead to intermediate values of relevant pathogen traits. I appreciate that the authors use a more operational definition of virulence, as the narrow definition of virulence as mortality has probably limited the applicability of evolutionary models to natural host-pathogen systems. And in both the transmission-virulence trade-off and the transmission-recovery trade-off, the key feature is that there is an underlying pathogen trait which simultaneously increases transmission rate and the rate that infections end, either due to host mortality (virulence) or host recovery. However, I found the invocation of the transmission-virulence trade-off to be a bit distracting because I kept thinking about how that trade-off is defined in theory, and because the authors' title emphasizes the transmission-recovery trade-off, not transmission-virulence. Moreover, many authors have argued that pathogen fitness should be highly sensitive to recovery (e.g. Frank & Schmid-Hempel 2008, Bull & Luring 2014), but transmission-recovery trade-offs are critically understudied (Read & Keeling 2006, Alizon 2008). I suppose I am making an appeal that, because transmission-recovery trade-offs are less appreciated, the authors could do more to use their results to argue that an emphasis on transmission-virulence trade-offs may overlook other important trade-offs that drive pathogen evolution. I also think that a focus on transmission-recovery trade-offs would be easier to write about, and would avoid the need to define virulence as peak viral load. Instead you could use peak viral load as the empirical measure of the pathogen trait that drives the trade-off, in the same way that Fraser and colleagues have done with HIV.

I was also wondering why the authors chose to focus on viral production rate, rather than viral infectivity rate, given their comment on lines 345-347, "that a significant portion of the variation in dengue viral load can be explained by differences in viral infectivity rate." Given that it is viral load that underlies both transmission and recovery, does it matter whether, e.g., the trait on the x-axis in Fig. 3a,b is "viral infectivity rate" or "viral production rate"? I realize that variation in viral infectivity generates the uncertainty around each of the lines, so I suppose I am wondering about the amount of variation in either trait. Was the focus on viral production rate because that shows more variation within a serotype, whereas variation in viral infectivity rate is due more to host variation and serotype variation? Or was it because the amount of variation in viral production is likely to be much larger than the variation in viral infectivity?

Minor comments:

Fig. 2c,d were generated using the parameterized within-host model – is the variation in peak viral load and clearance rate due to both measurement error (mentioned in the caption) and variation in viral infectivity rate (mentioned in the Discussion), or only to measurement error? Either way, do the authors have any idea why there is more variation in clearance rate than peak viral load?

Grammatical errors:

Line 186: "depends" rather than "depend"

Line 342: "reflecting" rather than "reflectivity"

Line 375: "accommodates measurements below the limit of detection" rather than "accommodates below the limit of detection measurements"

I think that the statistical analyses are both appropriate and valid, and I am confident that I and other researchers could reproduce the results.

Reviewer #2 (Remarks to the Author):

This paper explores the potential role of a trade-off between transmission and recovery rate in shaping the evolution of virulence in dengue virus. The paper presents a nice combination of (1) analysing clinical data, which shows a positive relationship between peak viral load (a proxy for virulence, also correlated with transmission) and viral clearance rate (recovery rate), (2) fitting a mechanistic within-host model to clinical data, which shows that this same relationship emerges due to feedbacks between viral growth and immune stimulation, (3) simulating a model that combines these within-host processes and between-host transmission, which shows that 'transmission potential' (a proxy for viral fitness) is maximised at intermediate peak viral loads and that secondary infections have lower peak loads than primary ones, and (4) an adaptive dynamics analysis that supports the inferences from (3): when two strains are circulating in a host population, and secondary infections are therefore possible, the evolutionary stable level of virulence is lower.

I was excited to read this paper and it didn't disappoint. It is an elegant combination of analyses and, despite the different approaches used, I found the methods section incredibly lucid. Indeed, seeing this paper in print will allow me to pass it out as a guide to students. The inferences and insights that emerge from this paper are interesting as well. I agree with the authors when they say that, currently, HIV is the only human disease for which a trade-off governing the evolution of intermediate viral virulence has been well-established and quantified (l. 6-7), so this study provides a nice, complementary example given that it is a different underlying trade-off that is implicated.

Despite (or in addition to) my general positivity about the paper, I have a number of questions and comments.

1. The use of "virulence" throughout the manuscript becomes a bit murky. It is stated early on that disease severity is a consequence of host traits (l. 29-31) and that it is correlated with viral load (l. 39-41). It is not clear to me whether the second clause has controlled for the relationship in the first clause. (Is this true, independent of whether it is a first or second infection?) Regardless, the text sort of gives the impression of viral load (the operational definition of virulence, l. 59) being a viral trait, when it is instead a consequence of both a viral trait (viral production rate) and host traits (i.e., immune responses). Obviously, this is implicit in the analyses, but without making it crystal clear in the text it becomes hard to, for example, hold in one's head that the optimal viral load for secondary infections is always lower, but the outcome (i.e., the actual viral load) in secondary infections is always higher. Analogously, understanding that secondary infections ARE more virulent and that this should lead to selection for lower virulence requires some similar mental gymnastics. I wonder if confusion can be avoided with more sparing use of the word "virulence", or more clarity early on (e.g., somewhere around line 33?) about viral load only being partially controlled by the virus.

2. The abstract notes "well-documented variation in dengue virus production rate" (as do lines 105-107),

but this parameter is fixed in the within-host model fitted to data. Instead, inter-individual variation is captured by variation in infectivity. Is there some rationale, in addition to non-identifiability, for fixing ω specifically? Given that the rest of the inferences flow from models parameterised in this first part -- while varying ω -- and that parameter estimates will (presumably?) be impacted by the choice of fixed parameters, it is not clear to me whether fitting or fixing ω makes the most sense. I am legitimately curious why the authors chose not to let ω be free here.

3. The big cloud of black in Fig 2a&b feels somewhat misleading. There appear to be some trajectories that are not following the model prediction, but it is hard to decipher independent trajectories. We would certainly expect this, given that the model is fit to all of the data at once, rather than individuals, and then the 'average' predicted trajectory is plotted. Still, I'd personally like to see the black lines plotted a lot thinner, to be able to decipher individual infections then and better gauge both variation and model fit.

4. In Figures 2c&d, I do not find the p-value reporting compelling since those could likely have been made even smaller by doubling the number of simulations. Is there a good case for reporting these? More convincing to me is that fact that the regression slopes are remarkably close to what was estimated for the data in Fig 1, despite the fact that the relationship isn't immediately obvious by eye. Incidentally, I would also suggest remaking these figures since the solid line is hidden by the open points.

5. My brain is having a hard time converting dynamics into fitness estimates. For example, in Figure S2b, it is clear that with a lower ω , the blue curve has a very slightly lower peak, but also lasts much longer. I wonder if both of these figures could be coupled with a plot of transmission potential (equation 3) over time? Would it be informative to see these plots of how fitness accrues over the course of infection?

6. Does the outcome of the evolutionary analyses with two serotypes circulating depend on the fraction of hosts who have a primary infection at equilibrium? If so, it would be good to know where we are in 'epidemiological variable' space with these outcomes and how much optimal production rate (and consequently, the difference in viral load between primary and secondary infections) could change. If there was some way of varying a between-host parameter that altered the 'predominance' of primary versus secondary infections, shifting the evolutionary optimum, this would be cool to see too and would provide some further support to the inferences in lines 213-215.

7. Given the discussion of virulence evolution depending on epidemiological context, can specific predictions be made about the epidemiological context in HCMC from what has been inferred about viral traits here? It would be very interesting to read some speculation about this and see how that fits with what is known about dengue transmission in the region. Figure 4b&c predict that with two serotypes circulating primary infections will have lower peak loads than secondary infections and that this difference appears to be about half a log. This is precisely the difference that is estimated from the mechanistic model (l. 406), so could we infer that the relative importance of primary and secondary infections in HCMC is roughly the same as what is captured in the model in 4b?

8. Knowing only very little about immunity, it is quite surprising to me that the per cell rate of killing of T cells is three orders of magnitude lower than for NK cells (l. 362). Should I be surprised that the innate response is much more effective?

Minor comments.

- Semantically, I find it funny to argue that a data subset "is more representative of all infections", when it is surely better representative of fewer infections. I take the point that the fits to the full

dataset will be skewed by the high viral load infections, but I might avoid that exact phrasing.

- Why are there are two limits of detection (Figure 2 caption and Methods, l. 300)? Are the data below the first limit (plotted in figures 2a&b) taken to be reliable? Do we learn anything from that top horizontal line, or can it be dropped for simplicity?
- l. 172 I think you mean something different than "viral magnitude", which I might take to mean the size of the virus (!?)
- Ref 70. I think there's a last name missing here.

Reviewer #3 (Remarks to the Author):

This is an interesting study that investigates an important question, namely, is there an inherent trade-off between magnitude of replication and rapidity of clearance by the immune system during a dengue virus infection of humans? The study leverages a dataset on viral replication dynamics collected by a different group of investigators and uses modeling to identify trade-offs and assess how such trade-offs may influence dengue virus evolution. Findings for the trade-off between magnitude and duration of viral replication largely confirm a relationship previously proposed for dengue replication in non-human primate hosts, but the finding that predominance of secondary infections may select for lower dengue virulence is unexpected and intriguing.

Major critiques

Introduction: The authors compare the trade-offs inherent in HIV replication (high magnitude of replication leads to rapid progression to AIDS, lower magnitude of replication leads to longer clinical latency) and dengue virus. However there is a key difference between HIV and dengue: untreated, HIV inevitably causes AIDS, whereas the great majority of dengue cases are asymptomatic. Moreover, AIDS is a rather distinct disease, whereas dengue disease encompasses a broad spectrum of severity, indeed many experts reject the DF/DHF dichotomy. Here, the authors define DHF as "severe disease", but they never really specify what they mean by "severe". All of the patients from whom the empirical data were collected were in the clinic; that is one definition of severe disease. These concepts deserve a more nuanced consideration than they receive in this manuscript.

Introduction: The authors should also consider White et al. (2016, PLOS Biology, Variation in relapse frequency and the transmission potential of *Plasmodium vivax* malaria) in their list of human pathogens for which a virulence-transmission trade-off has been documented.

Results: The authors note that they have ignored potential trade-offs between replication in the human host and replication in the mosquito. They justify this by stating that there is little data on such associations. In so doing, they ignore a rather large body of literature on competitive displacements among dengue strains, which in many cases shows that a displacing strain is both more infectious for mosquitoes (under experimental conditions) and causes higher levels of disease in humans. In particular I am thinking of work from the Rico-Hesse, Harris and Hanley labs. Rather than just calling for more research, the authors should explicitly consider how such linked phenotypes (high levels of replication in humans and high infectivity for mosquitoes even when titers are held constant) would affect the outcome of their models.

Results: Because of their focus on trade-offs in human pathogens, the authors neglect to place their findings in the context of the broader literature and theory on evolution of virulence. Thus the paper may be of limited interest to those outside of the dengue/arbovirus research community.

Methods: It would be a great deal easier to read this section if the definition of each model parameter

and various estimates for parameter values were summarized in a table rather than folded into the text.

Methods: Simulating asymptomatic infections but just lowering viral loads makes the assumption that viral dynamics are otherwise similar in symptomatic and asymptomatic infections. Is there support for this assumption?

Methods: How sensitive are model results to the assumption that heterotypic cross protection lasts two years? This estimate is rather controversial.

Minor critiques

Throughout- please change copies/ml to genome copies/ml to be clear about what is being measured

Line 6: change "in humans" to "among human pathogens"

Line 15: The diversity of dengue virus can be clustered in many ways- genotypes are arbitrarily defined as differing by 6% nt divergence- add this information.

Line 22: Add "at least in part" before "is due to"

Line 25: A process does not propose; a person or hypothesis does so.

Fig 1 legend: This sentence is a bit confusing: The relationship is significantly positive at the $p = 0.1$ level (slope $m = 0.3$; $p = 0.01$). I got it after 2 reads but initially it looks like a typo.

Figure 4: symbols are so small that this figure is really hard to read

Discussion, 1st paragraph. It is worth noting that reference 42 also invoked immune clearance as the mechanism for the trade-off between magnitude and duration of viral replication.

Methods: In the Vietnam study from which the empirical data were drawn, is it known whether the serotypes encompassed only a single genotype or multiple genotypes?

Line 401: ADE is a hypothesis- the term theory is reserved for established mechanisms like gravity and evolution.

Methods: The "virulence evolution" section is confusing because of the term "invading serotype". Is this a different serotype or a new genotype of the established serotype? Clarify.

Figure S6 legend is rather confusing as written.

Reviewer #1 (Remarks to the Author):

In this study, Ben-Shachar and Koelle provide strong evidence for the existence of a key fitness trade-off that promotes the evolution of intermediate virulence in human dengue. Given the paucity of studies documenting such fitness trade-offs, relative to their importance in mathematical theory for pathogen evolution, this is a very important and novel study. Moreover, the study is comprehensive; I particularly appreciated the analysis of pathogen evolution across different epidemiological contexts (primary vs. secondary infection). The manuscript is very well-written, and the Discussion section touched on all of the major points that I wanted to see. In my opinion, this is an important study that will appeal to a broad audience, and thus warrants publication in Nature Communications.

My main comment is one that I am sure the authors considered in drafting the manuscript: why structure the Introduction and Discussion around transmission-virulence trade-off theory, when the data (and title) clearly supports a transmission-recovery trade-off? Both trade-offs have been shown to lead to intermediate values of relevant pathogen traits. I appreciate that the authors use a more operational definition of virulence, as the narrow definition of virulence as mortality has probably limited the applicability of evolutionary models to natural host-pathogen systems. And in both the transmission-virulence trade-off and the transmission-recovery trade-off, the key feature is that there is an underlying pathogen trait which simultaneously increases transmission rate and the rate that infections end, either due to host mortality (virulence) or host recovery. However, I found the invocation of the transmission-virulence trade-off to be a bit distracting because I kept thinking about how that trade-off is defined in theory, and because the authors' title emphasizes the transmission-recovery trade-off, not transmission-virulence. Moreover, many authors have argued that pathogen fitness should be highly sensitive to recovery (e.g. Frank & Schmid-Hempel 2008, Bull & Luring 2014), but transmission-recovery trade-offs are critically understudied (Read & Keeling 2006, Alizon 2008). I suppose I am making an appeal that, because transmission-recovery trade-offs are less appreciated, the authors could do more to use their results to argue that an emphasis on transmission-virulence trade-offs may overlook other important trade-offs that drive pathogen evolution. I also think that a focus on transmission-recovery trade-offs would be easier to write about, and would avoid the need to define virulence as peak viral load. Instead you could use peak viral load as the empirical measure of the pathogen trait that drives the trade-off, in the same way that Fraser and colleagues have done with HIV.

We appreciate this thoughtful comment, and agree that emphasizing the transmission-clearance trade-off we found for dengue virus would strengthen the manuscript considerably. We have reworked the introduction to more clearly indicate that the trade-off we are finding in dengue virus involves viral clearance, rather than pathogen-induced host mortality. We have further gone through the manuscript and removed explicit reference to the transmission-virulence trade-off, as the term virulence in the context of this trade-off is largely synonymous with disease-induced mortality rates (which we do not consider in this manuscript). Removal of this term now allows us to more effectively argue that evolution of dengue virulence, as defined here by the probability of developing severe disease, is impacted by the trade-off between dengue transmission and the virus's clearance rate. We now also cite the two papers listed above that we had not already previously cited (Frank & Schmid-Hempel 2008 and Read & Keeling 2006),

both of which support the notion that pathogen clearance rates are likely to be important fitness components structuring the evolution of pathogen virulence.

I was also wondering why the authors chose to focus on viral production rate, rather than viral infectivity rate, given their comment on lines 345-347, “that a significant portion of the variation in dengue viral load can be explained by differences in viral infectivity rate.” Given that it is viral load that underlies both transmission and recovery, does it matter whether, e.g., the trait on the x-axis in Fig. 3a,b is “viral infectivity rate” or “viral production rate”? I realize that variation in viral infectivity generates the uncertainty around each of the lines, so I suppose I am wondering about the amount of variation in either trait. Was the focus on viral production rate because that shows more variation within a serotype, whereas variation in viral infectivity rate is due more to host variation and serotype variation? Or was it because the amount of variation in viral production is likely to be much larger than the variation in viral infectivity?

We focus on dengue virus evolving its viral production rate because there is extensive literature documenting that variation in this viral phenotype has a viral genetic basis (rather than a host factor basis), and because natural variation in these viral genotypes/phenotypes exists. When we look at the viremia data from Ho Chi Minh City, we fit a model that allows for variation in the viral infectivity rate, however. This is because there are substantial differences in viral load dynamics between individuals, and these differences are thought to arise primarily from host factors, such as previous exposure histories and host genetic factors. We think of the viremia data as capturing a snapshot in time (in 2007/2008, when the data were collected), with little variation in the viral production rate of the circulating strains and significant variation in dengue virus infectivity rates due to host heterogeneity in the population. We then consider evolution of dengue viral production rates over time in the context of host heterogeneity (captured by variation in viral infectivity rates). We added text to the Introduction and to the Methods section to clarify what within-host parameters predominantly reflect host factors versus viral factors, indicating that we can largely think of dengue virus production rates as being a viral factor and dengue virus infectivity rates as being a host factor.

Minor comments:

Fig. 2c,d were generated using the parameterized within-host model – is the variation in peak viral load and clearance rate due to both measurement error (mentioned in the caption) and variation in viral infectivity rate (mentioned in the Discussion), or only to measurement error? Either way, do the authors have any idea why there is more variation in clearance rate than peak viral load?

Variation is due to both measurement error and variation in the viral infectivity rate, and we now make this clear in the figure legend. There is more variation in the viral clearance rate than in the viral peak because variation in the viral infectivity rate specifically affects viral clearance more than peak viral load (when shown on the log₁₀ scale).

Grammatical errors:

Line 186: “depends” rather than “depend”

Line 342: “reflecting” rather than “reflectivity”

Line 375: “accommodates measurements below the limit of detection” rather than “accommodates below the limit of detection measurements”

We have fixed these errors.

I think that the statistical analyses are both appropriate and valid, and I am confident that I and other researchers could reproduce the results.

Reviewer #2 (Remarks to the Author):

This paper explores the potential role of a trade-off between transmission and recovery rate in shaping the evolution of virulence in dengue virus. The paper presents a nice combination of (1) analysing clinical data, which shows a positive relationship between peak viral load (a proxy for virulence, also correlated with transmission) and viral clearance rate (recovery rate), (2) fitting a mechanistic within-host model to clinical data, which shows that this same relationship emerges due to feedbacks between viral growth and immune stimulation, (3) simulating a model that combines these within-host processes and between-host transmission, which shows that ‘transmission potential’ (a proxy for viral fitness) is maximised at intermediate peak viral loads and that secondary infections have lower peak loads than primary ones, and (4) an adaptive dynamics analysis that supports the inferences from (3): when two strains are circulating in a host population, and secondary infections are therefore possible, the evolutionary stable level of virulence is lower.

I was excited to read this paper and it didn't disappoint. It is an elegant combination of analyses and, despite the different approaches used, I found the methods section incredibly lucid. Indeed, seeing this paper in print will allow me to pass it out as a guide to students. The inferences and insights that emerge from this paper are interesting as well. I agree with the authors when they say that, currently, HIV is the only human disease for which a trade-off governing the evolution of intermediate viral virulence has been well-established and quantified (l. 6-7), so this study provides a nice, complementary example given that it is a different underlying trade-off that is implicated.

Despite (or in addition to) my general positivity about the paper, I have a number of questions and comments.

1. The use of "virulence" throughout the manuscript becomes a bit murky. It is stated early on that disease severity is a consequence of host traits (l. 29-31) and that it is correlated with viral load (l. 39-41). It is not clear to me whether the second clause has controlled for the relationship in the first clause. (Is this true, independent of whether it is a first or second infection?) Regardless, the text sort of gives the impression of viral load (the operational definition of virulence, l. 59) being a viral trait, when it is instead a consequence of both a viral trait (viral production rate) and host traits (i.e., immune responses). Obviously, this is implicit in the analyses, but without making it crystal clear in the text it becomes hard to, for example, hold in one's head that the optimal viral load for secondary infections is always lower, but the outcome (i.e., the actual viral load) in secondary infections is always higher. Analogously, understanding

that secondary infections ARE more virulent and that this should lead to selection for lower virulence requires some similar mental gymnastics. I wonder if confusion can be avoided with more sparing use of the word "virulence", or more clarity early on (e.g., somewhere around line 33?) about viral load only being partially controlled by the virus.

We greatly appreciate this comment from the reviewer. First, to address the reviewer's question about whether disease severity is correlated with viral load, once host immune status has been accounted for: yes, higher peak viral load is associated with higher risk of developing severe dengue disease, when considering only primary infections and also only secondary infections (Vaughn et al 2000, Tricou et al 2011). This makes sense given that the viral infection drives an immune response, and that dengue disease comes about through immunopathology.

In our attempt to eliminate any mental gymnastics, we have added text to the Introduction and Results sections that clearly states that viral load is jointly determined by viral traits (such as the production rate that can differ by genotype and serotype) and host traits (such as host immune status and host genetics). In response to both this reviewer and reviewer #1, we also now no longer allude to a transmission-virulence trade-off, but focus exclusively on the transmission-clearance tradeoff. We note that where on this trade-off range the virus lies has implications for viral load dynamics and therefore also virulence, as defined as the probability of an infected individual developing severe dengue disease.

2. The abstract notes "well-documented variation in dengue virus production rate" (as do lines 105-107), but this parameter is fixed in the within-host model fitted to data. Instead, inter-individual variation is captured by variation in infectivity. Is there some rationale, in addition to non-identifiability, for fixing ω specifically? Given that the rest of the inferences flow from models parameterised in this first part -- while varying ω -- and that parameter estimates will (presumably?) be impacted by the choice of fixed parameters, it is not clear to me whether fitting or fixing ω makes the most sense. I am legitimately curious why the authors chose not to let ω be free here.

The well-documented variation in dengue virus production rates comes from either spatially distinct regions or from studies that have considered different dengue genotypes where one genotype has rapidly replaced a circulating genotype. The variation that has been documented implies to us that dengue virus has the genetic capacity to modify this viral trait. The variation that is observed empirically does not come from variation present in a single location at a single timepoint, and we thus think of the viral production rate ω as being constant when we fit it to data. We have added text to clarify this point in the Methods section that describes the fitting procedure.

3. The big cloud of black in Fig 2a&b feels somewhat misleading. There appear to be some trajectories that are not following the model prediction, but it is hard to decipher individual trajectories. We would certainly expect this, given that the model is fit to all of the data at once, rather than individuals, and then the 'average' predicted trajectory is plotted. Still, I'd personally like to see the black lines plotted a lot thinner, to be able to decipher individual infections then and better gauge both variation and model fit.

We have made the black lines thinner in Figures 2A and 2B. With these plots, it is difficult to show variation in model predictions and how these predictions look plotted against the data. This is because, under any model parameterization, to plot an individual's viral load as a function of time since infection, an incubation period needs to be estimated for that individual (see Methods). This incubation period quantifies the period of time from infection to the onset of symptoms. In Figures 2A and 2B, we do not show individual fits when the viral infectivity is ± 2 standard deviations of the mean for the sake of clarity. Below, for the reviewer, we include this plot for Figure 2A. Individual trajectories, offset by their estimated incubation periods, are shown in gray, along with model predictions using ± 2 standard deviations from the viral infectivity mean.

4. In Figures 2c&d, I do not find the p-value reporting compelling since those could likely have been made even smaller by doubling the number of simulations. Is there a good case for reporting these? More convincing to me is that fact that the regression slopes are remarkably close to what was estimated for the data in Fig 1, despite the fact that the relationship isn't immediately obvious by eye. Incidentally, I would also suggest remaking these figures since the solid line is hidden by the open points.

We have remade Figures 2C and 2D so the points do not obscure the regression lines. In the figure legend, we now also emphasize the slope of the lines, rather than the p-values. We have further modified Figures 1A and 1B to similarly emphasize the slope over the p-values.

5. My brain is having a hard time converting dynamics into fitness estimates. For example, in Figure S2b, it is clear that with a lower ω , the blue curve has a very slightly lower peak, but also lasts much longer. I wonder if both of these figures could be coupled with a plot of transmission potential (equation 3) over time? Would it be informative to see these plots of how fitness accrues over the course of infection?

We have added an additional subplot to Fig. S2 (Fig. S2B) that shows how transmission potential accrues over time since infection for primary infections. We have removed the secondary infection simulations, since the trade-offs are already clearly shown for the primary model simulations, and we wanted to try to minimize the amount of supplemental figures and subplots without loss of content.

6. Does the outcome of the evolutionary analyses with two serotypes circulating depend on the fraction of hosts who have a primary infection at equilibrium? If so, it would be good to know where we are in 'epidemiological variable' space with these outcomes and how much optimal production rate (and consequently, the difference in viral load between primary and secondary infections) could change. If there was some way of varying a between-host parameter that altered the 'predominance' of primary versus secondary infections, shifting the evolutionary optimum, this would be cool to see too and would provide some further support to the inferences in lines 213-215.

Yes, the reviewer is absolutely correct and the outcome of the evolutionary analysis does depend on the fraction of hosts who have a primary infection at equilibrium. (In the single serotype scenario, all hosts are experiencing a primary infection and the outcome is effectively just determined by the relative basic reproduction numbers of the strains under different viral production rates.) To emphasize that the evolutionary analysis depends on the fraction of hosts who have a primary infection at equilibrium, we have removed our previous Figure 4B, which showed the pairwise invasibility plot for the 2-circulating serotypes case. In its place, we now instead show a subplot which has the parameter f on the x-axis, quantifying the factor by which the transmission potential is scaled to capture the process of epidemiological transmission. The factor f is the product of mosquito-to-human population sizes, the square of the mosquito biting rate, and the probability that an infected mosquito transmits dengue to a human upon a bite (see Methods). As such it can be thought of as scaling the contact rate between humans and the vector. In Figure 4B, we show what the evolutionary stable viral production rate is along this continuum of transmission intensity f . As we would expect from our original analysis, the evolutionary stable viral production rate is always lower when two serotypes are endemic compared to when a single serotype is circulating. However, there are differences in evolutionary outcomes as a function of f , and these (interestingly) depend on the type and duration of heterologous cross-protection assumed. (We consider different durations and types of cross-protection in response to reviewer #3's comments – see below.)

7. Given the discussion of virulence evolution depending on epidemiological context, can specific predictions be made about the epidemiological context in HCMC from what has been inferred about viral traits here? It would be very interesting to read some speculation about this and see how that fits with what is known about dengue transmission in the region. Figure 4b&c predict that with two serotypes circulating primary infections will have lower peak loads than secondary infections and that this difference appears to be about half a log. This is precisely the difference that is estimated from the mechanistic model (l. 406), so could we infer that the relative importance of primary and secondary infections in HCMC is roughly the same as what is captured in the model in 4b?

While we think that the reviewer's comments and suggestions here are highly interesting, we would like to refrain from comparing the model predictions from Figure 4B to HCMC's epidemiological context. This is because there are four serotypes endemically circulating in HCMC, and we do not know enough about post-secondary infections to be able to meaningfully apply our evolutionary analyses to HCMC. Specifically, as elaborated on in the Discussion, very little is known about viral load dynamics in post-secondary infections, the processes that regulate these dynamics, and the role of post-secondary infections in onward transmission.

8. Knowing only very little about immunity, it is quite surprising to me that the per cell rate of killing of T cells is three orders of magnitude lower than for NK cells (l. 362). Should I be surprised that the innate response is much more effective?

As we detailed in the Methods section, the killing rates of both NK cells and T-cells are either structurally or practically unidentifiable without specified initial conditions (and to a lesser extent death rates). The per-cell rate of killing of NK-cells was set arbitrarily because this rate and the NK-cell activation rate are structurally unidentifiable (Ben-Shachar et al 2016). Our choices of death rates and initial conditions were based on (limited) empirical data. We have also previously shown that the initial number of T-cells and the per-cell rate of killing of T-cells are practically unidentifiable. We therefore set the initial number of T-cells based on reasonable estimates in (de Matos 2015). To determine a value at which to set to rate of T-cell killing, we first fixed the rate of killing of T-cells to various values and estimated the T-cell activation rate. We then ended up setting the rate of T-cell killing to 10^{-6} per day so that T-cell counts would reach maximum values on the order of 10^6 cells/day, consistent with empirical data (de Matos 2015). That said, without immune data for these patients, it would be unwise for us to conclude anything about the true relative magnitudes of T-cell versus NK cell killing rates, as it is even hard to separate the role of the innate and the adaptive immune response in secondary infections (as described in the Methods section).

Minor comments.

- Semantically, I find it funny to argue that a data subset "is more representative of all infections", when it is surely better representative of fewer infections. I take the point that the fits to the full dataset will be skewed by the high viral load infections, but I might avoid that exact phrasing.

We agree with the reviewer and have modified this text.

- Why are there are two limits of detection (Figure 2 caption and Methods, l. 300)? Are the data below the first limit (plotted in figures 2a&b) taken to be reliable? Do we learn anything from that top horizontal line, or can it be dropped for simplicity?

There were two different assays used for the measurement of viremia in the original study (Clapham et al 2014). In Figure 2, we therefore show the two different limits of detection, one for each assay. We cannot simply drop the top horizontal line because some of the viremia measurements fell below this limit of detection, rather than the lower one.

- l. 172 I think you mean something different than "viral magnitude", which I might take to mean the size of the virus (!?)

We have fixed this awkward wording. We meant peak viremia.

- Ref 70. I think there's a last name missing here.

We have corrected this reference.

Reviewer #3 (Remarks to the Author):

This is an interesting study that investigates an important question, namely, is there an inherent trade-off between magnitude of replication and rapidity of clearance by the immune system during a dengue virus infection of humans? The study leverages a dataset on viral replication dynamics collected by a different group of investigators and uses modeling to identify trade-offs and assess how such trade-offs may influence dengue virus evolution. Findings for the trade-off between magnitude and duration of viral replication largely confirm a relationship previously proposed for dengue replication in non-human primate hosts, but the finding that predominance of secondary infections may select for lower dengue virulence is unexpected and intriguing.

Major critiques

Introduction: The authors compare the trade-offs inherent in HIV replication (high magnitude of replication leads to rapid progression to AIDS, lower magnitude of replication leads to longer clinical latency) and dengue virus. However there is a key difference between HIV and dengue: untreated, HIV inevitably causes AIDS, whereas the great majority of dengue cases are asymptomatic. Moreover, AIDS is a rather distinct disease, whereas dengue disease encompasses a broad spectrum of severity, indeed many experts reject the DF/DHF dichotomy. Here, the authors define DHF as "severe disease", but they never really specify what they mean by "severe". All of the patients from whom the empirical data were collected were in the clinic; that is one definition of severe disease. These concepts deserve a more nuanced consideration than they receive in this manuscript.

We have clarified throughout the manuscript that the data we use to fit the model includes symptomatic cases of dengue (DF and DHF). We refer to DHF as severe dengue disease based on the 1997 WHO guidelines on symptomatic classification of dengue. Other dengue classifications include dengue with and without warning signs as well as clinical management scores. However, we feel that DF/DHF classification (from the 1997 WHO guidelines) is still the most frequently used classification, and, importantly, is the classification scheme that was used to classify disease in the HCMC dataset. Furthermore, a recent paper has shown large overlap between these three dengue classification schemes (see Table S5 in Katzelnick et al. (2017) *Science*).

Introduction: The authors should also consider White et al. (2016, PLOS Biology, Variation in relapse frequency and the transmission potential of *Plasmodium vivax* malaria) in their list of human pathogens for which a virulence-transmission trade-off has been documented.

We thank the reviewer for pointing us towards this paper, which shows that malaria's transmission potential is optimized at intermediate relapse frequencies. The results presented in this paper are very interesting, but we feel that they can only loosely be brought to bear on the transmission-clearance trade-off we find evidence for (since relapse frequency is related - but still rather distinct from - the rate of clearance from an acute infection). Given the first two reviewers' comments, which underscore the importance of clearly specifying the trade-offs we have found for dengue, we have decided to not cite this paper, as it would introduce yet another trade-off type into the mix, and one that is not specifically at play in dengue. We do, however, now also cite malaria as another human pathogen for which data exist that support the assumptions of evolution of virulence theory (Mackinnon & Read (2004)).

Results: The authors note that they have ignored potential trade-offs between replication in the human host and replication in the mosquito. They justify this by stating that there is little data on such associations. In so doing, they ignore a rather large body of literature on competitive displacements among dengue strains, which in many cases shows that a displacing strain is both more infectious for mosquitoes (under experimental conditions) and causes higher levels of disease in humans. In particular I am thinking of work from the Rico-Hesse, Harris and Hanley labs. Rather than just calling for more research, the authors should explicitly consider how such linked phenotypes (high levels of replication in humans and high infectivity for mosquitoes even when titers are held constant) would affect the outcome of their models.

We thank the reviewer for his/her suggestion to better acknowledge existing research showing a relationship between the viral replication rate in the human host and viral replication in the mosquito. In the Discussion section, we now allude to several papers that have found evidence for dengue strains with high replication rates in humans also having a replication advantage in mosquitoes. Specifically, Hanley et al. (2008) showed that an invasive DENV-3 strain disseminated more efficiently throughout a mosquito, with the implication being that more efficient dissemination would increase transmission probability to a susceptible host. We also cite Cologna et al. (2005), which found that virulent dengue strains have higher viral replication rates in the mosquito as well as in humans. Finally, we cite OhAinle et al. (2011), which showed that a clade of DENV-2, which was associated with higher viremia, had higher fitness in both mosquito and human cell lines. Unfortunately, in none of these cases is it known whether greater replication ability or dissemination in the mosquito would lead to more successful transmission from infected mosquito to susceptible human. Under the assumption that higher levels of viremia or dissemination in the mosquito leads to higher transmission success to humans, these findings would simply shift the optimal viral production rates to higher levels. In the Discussion, we point the reader to Note S3 and Figure S8, which implement this possible effect.

Results: Because of their focus on trade-offs in human pathogens, the authors neglect to place their findings in the context of the broader literature and theory on evolution of virulence. Thus the paper may be of limited interest to those outside of the dengue/arbovirus research community.

We have added text to the introduction and results section to place our findings more broadly into the evolution of virulence literature, particularly in the context of demonstrating the presence of a transmission-clearance trade-off.

Methods: It would be a great deal easier to read this section if the definition of each model parameter and various estimates for parameter values were summarized in a table rather than folded into the text.

We have added a table (Supplementary Table 1) to the Supplemental Material. This table describes each model parameter and provides its statistical estimates or assigned value.

Methods: Simulating asymptomatic infections but just lowering viral loads makes the assumption that viral dynamics are otherwise similar in symptomatic and asymptomatic infections. Is there support for this assumption?

We implicitly incorporate asymptomatic infections into our analysis by analyzing the viral load data with lower peak viral loads. We believe this is a fair assumption for two reasons: (1) a study that quantified viral load in asymptomatic and symptomatic individuals found that viral load measurements were generally lower for asymptomatic individuals compared to symptomatic individuals (Duong et al. (2015)). (2) We expect peak viral load to be lower for asymptomatic infections as peak viral load has been shown to be correlated with severe disease (Vaughn et al. (2000), Tricou et al. (2011)). Data on the duration of infection for asymptomatic infections are, to the best of our knowledge, unfortunately not available. The reviewer is therefore correct in that, by analyzing the subset of data with individuals with low viral loads, we are assuming that any trade-offs between peak viral load and viral clearance that are observed in symptomatic infections with lower peak viral loads extend to asymptomatic infections. While there is no strong empirical support for this assumption, there are also no data indicating otherwise.

Methods: How sensitive are model results to the assumption that heterotypic cross protection lasts two years? This estimate is rather controversial.

In Figure 4B, we now show how the transmission intensity of dengue (quantified by the parameter f) affects optimal viral production rates. We show this for three different cases: under the assumption of no heterotypic cross-protection (0 years cross-protection), under the assumption of 2 years of heterotypic cross-protection of the ‘classical’ type, and under the assumption of 2 years of heterotypic cross-protection of the ‘clinical’ type. (Please see Nagao & Koelle (2008) *PNAS* for the definition of ‘classical’ vs ‘clinical’ cross-protection.) From this figure, it is clear that, while the type and duration of heterotypic cross-protection does impact optimal viral production rates, the general conclusion that epidemiological scenarios with 2 circulating serotypes always select for lower viral production rates than an epidemiological scenario with only a single circulating serotype.

Minor critiques

Throughout- please change copies/ml to genome copies/ml to be clear about what is being measured

We have made these changes.

Line 6: change “in humans” to “among human pathogens”

We have made this change.

Line 15: The diversity of dengue virus can be clustered in many ways- genotypes are arbitrarily defined as differing by 6% nt divergence- add this information.

We have added this information.

Line 22: Add “at least in part” before “is due to”

This has been added.

Line 25: A process does not propose; a person or hypothesis does so.

This text has been modified.

Fig 1 legend: This sentence is a bit confusing: The relationship is significantly positive at the $p = 0.1$ level (slope $m = 0.3$; $p = 0.01$). I got it after 2 reads but initially it looks like a typo.

We have clarified this text.

Figure 4: symbols are so small that this figure is really hard to read

We have enlarged the symbols.

Discussion, 1st paragraph. It is worth noting that reference 42 also invoked immune clearance as the mechanism for the trade-off between magnitude and duration of viral replication.

We now note this.

Methods: In the Vietnam study from which the empirical data were drawn, is it known whether the serotypes encompassed only a single genotype or multiple genotypes?

Only partial genotype information was available for DENV-1: 130 out of the 147 DENV-1 infections belonged to genotype 1. For the other serotypes, no genotypic data were available.

Line 401: ADE is a hypothesis- the term theory is reserved for established mechanisms like gravity and evolution.

We have modified the text.

Methods: The “virulence evolution” section is confusing because of the term “invading serotype”. Is this a different serotype or a new genotype of the established serotype? Clarify.

We have clarified this by substituting the word ‘strain’ for ‘serotype’, since we are considering a new phenotype of an established serotype.

Figure S6 legend is rather confusing as written.

We have attempted to clarify Figure S6’s legend text.

REVIEWERS' COMMENTS:

Reviewer #1 (Remarks to the Author):

The authors have done an outstanding job addressing my previous comments, and, in my opinion, the comments of the other reviewers. I remain convinced that this is an important study, well worth publishing in Nature Communications. I have only one small comment.

The second paragraph of the discussion would seem to imply that only studies that have broadened the definition of virulence beyond host mortality have considered fitness trade-offs other than between transmission and virulence. That is not precisely correct, as one of the earliest studies showing that fitness peaks at intermediate virulence (Anderson and May's classic 1982 paper, "Coevolution of hosts and parasites") actually considered a trade-off between virulence and recovery. However, that was based on a simple analysis of R_0 , not the more sophisticated adaptive dynamics approach you undertake here.

Reviewer #2 (Remarks to the Author):

I appreciate the authors' thoughtful responses to the first round of comments and I am happy with their edits and replies. I have a few additional comments, all of which are relatively minor, and I'd be happy to leave it up to the judgement of the authors whether or not they should be addressed.

I think Reviewer 1's main comment may have pinpointed (in a much more lucid way!) why I was struggling with the word "virulence" in the first round of review. I like the new focus of the paper on transmission-recovery tradeoffs. Although the authors do a good job of equating virulence with disease severity early on, the remaining "virulence"s sprinkled throughout still slowed me down a bit and could be problematic for some readers. (It is hard not to instinctively conjure up an Anderson & May style R_0 expression, think about a tradeoff between two of the rates in that expression and then get confused about that tradeoff explaining the evolution of a third rate. Obviously, that's an incorrect way to think about it.) The authors should definitely do what they like here -- my brain gets to the right place in the end -- but I wonder if there could be gains in clarity by sticking with the more vague "disease severity" or the more precise, peak viral load (as suggested by that reviewer)?

I like the new Figures 4B and S2B, though I wonder if the y-axis on the latter should read "cumulative transmission potential"?

Finally, this version felt more dense than the first one, undoubtedly a partial consequence of the thoughtful changes made following review. I've flagged a few specific bits that could be edited.

- lines 59-65. I felt that more was said about results here than is necessary.
- l. 331 "a difficult task because of the difficulty" and l. 334-336 "Latin America is especially good...because countries in Latin America" felt a bit lumbering.
- l. 366-368. It's hard to parse any differences between these sentences.
- l.375-381. Somehow, it took me many tries to understand that the "alternative analysis" wasn't an alternative to something else presented here. Is it necessary to dwell on the previous analysis?

Reviewer #3 (Remarks to the Author):

The authors have done an excellent job of addressing all of the reviewer comments. The manuscript represents a rigorous study of broad importance and interest.

I have only minor suggestions to correct wording:

Minor points:

Line 22: Add result prior to "in severe disease"

Line 61: Change "developing" to "triggering"

Line 78: Change "comes about through" to "arises from"

Line 108-110: rather awkward sentence structure

Line 121: delete "globally sampled"

Line 129: Is the word "epidemiological" necessary here?

Line 414: Should this be "strain" rather than "serotype"?

Reviewer #1 (Remarks to the Author):

The authors have done an outstanding job addressing my previous comments, and, in my opinion, the comments of the other reviewers. I remain convinced that this is an important study, well worth publishing in Nature Communications. I have only one small comment.

The second paragraph of the discussion would seem to imply that only studies that have broadened the definition of virulence beyond host mortality have considered fitness trade-offs other than between transmission and virulence. That is not precisely correct, as one of the earliest studies showing that fitness peaks at intermediate virulence (Anderson and May's classic 1982 paper, "Coevolution of hosts and parasites") actually considered a trade-off between virulence and recovery. However, that was based on a simple analysis of R_0 , not the more sophisticated adaptive dynamics approach you undertake here.

We assume that the reviewer is referring to the Introduction, and not the Discussion here. We agree with the reviewer's comment, and have added text to the end of the first paragraph indicating that the evolution of virulence literature has also considered the trade-off between virulence (as defined as disease-induced mortality) and recovery rate (citing this 1982 reference):

“While a trade-off between transmission rate and disease-induced mortality is the classical trade-off considered in evolution of virulence theory, early work has also considered alternative trade-offs, for example, between the rate of recovery and the rate of disease-induced mortality (Anderson and May, 1982).”

Reviewer #2 (Remarks to the Author):

I appreciate the authors' thoughtful responses to the first round of comments and I am happy with their edits and replies. I have a few additional comments, all of which are relatively minor, and I'd be happy to leave it up to the judgement of the authors whether or not they should be addressed.

I think Reviewer 1's main comment may have pinpointed (in a much more lucid way!) why I was struggling with the word “virulence” in the first round of review. I like the new focus of the paper on transmission-recovery tradeoffs. Although the authors do a good job of equating virulence with disease severity early on, the remaining “virulence”s sprinkled throughout still slowed me down a bit and could be problematic for some readers. (It is hard not to instinctively conjure up an Anderson & May style R_0 expression, think about a tradeoff between two of the rates in that expression and then get confused about that tradeoff explaining the evolution of a third rate. Obviously, that's an incorrect way to think about it.) The authors should definitely do what they like here -- my brain gets to the right place in the end -- but I wonder if there could be gains in clarity by sticking with the more vague “disease severity” or the more precise, peak viral load (as suggested by that reviewer)?

We have re-read the manuscript again in its entirety and have replaced instances of “virulence” with “disease severity” or “peak viral load” when we thought it would clarify the text. However,

in some places, we kept the word “virulence” to ensure that this work’s relevance to the larger body of work on virulence evolution remains intact.

I like the new Figures 4B and S2B, though I wonder if the y-axis on the latter should read “cumulative transmission potential”?

The y-axis label has been changed.

Finally, this version felt more dense than the first one, undoubtedly a partial consequence of the thoughtful changes made following review. I’ve flagged a few specific bits that could be edited.

- lines 59-65. I felt that more was said about results here than is necessary.

We have shortened this section, and left details to the Results section.

- l. 331 “a difficult task because of the difficulty” and l. 334-336 “Latin America is especially good...because countries in Latin America” felt a bit lumbering.

These lines have been rephrased.

- l. 366-368. It’s hard to parse any differences between these sentences.

We have taken out the following redundant sentence: “We considered the relationships between maximum viral load and viral clearance rate using only this subset of individuals.”

- l. 375-381. Somehow, it took me many tries to understand that the “alternative analysis” wasn’t an alternative to something else presented here. Is it necessary to dwell on the previous analysis?

We have removed this text for clarity.

Reviewer #3 (Remarks to the Author):

The authors have done an excellent job of addressing all of the reviewer comments. The manuscript represents a rigorous study of broad importance and interest.

I have only minor suggestions to correct wording:
Minor points:

Line 22: Add result prior to “in severe disease”

This has been changed.

Line 61: Change “developing” to “triggering”

This has been changed.

Line 78: Change “comes about through” to “arises from”

This has been changed.

Line 108-110: rather awkward sentence structure

This has been changed to “In order to consider asymptomatic infections, we re-fit the model to a subset of the data consisting of patients with lower viral loads (Methods).”

Line 121: delete “globally sampled”

This has been deleted.

Line 129: Is the word “epidemiological” necessary here?

We have taken out the word “epidemiological”.

Line 414: Should this be “strain” rather than “serotype”?

In our previous study we looked at differences in serotype, not strain.

Specific Editorial Requests:

*The following Methods subheadings are too long; please shorten to maximum of 60 characters including spaces.

“Within-host model parameterization using the full HCMC dataset”

“Within-host model parameterization using a subset of the HCMC dataset”

These have been shortened.

*We do not allow multiple levels of subheadings. Please alter sections so that the sub-subheadings under “Quantifying population-level viral transmission potential” are either separate sections or regular paragraphs.

We have made the heading below “quantifying population-level viral transmission potential” subheadings.

*After the ‘Author Contributions’ statement, please add a Competing Interests statement indicating any competing financial or other interests.

This has been added.

*Please upload individual figure files with the next submission.

This has been added.

* In the figure legends (Figs 1, 2) please indicate the name of the statistical test and any relevant test statistics wherever p-values are given.

This has been added.

* In the SI file, please rename “References” as “Supplementary References”.

This has been changed.

* Your paper will be accompanied by a two-sentence editor's summary, of between 250-300 characters, when it is published on our homepage. Could you please approve the draft summary below or provide us with a suitably edited version in the cover letter.

"Theory predicts that pathogens are selected for intermediate virulence, but this is rarely tested empirically with human pathogens. Here, the authors show that dengue virus dynamics are mediated by a trade-off between transmission and clearance rates."

We include a modified version of this editor's summary in the cover letter. The modified summary is:

"Theory predicts that pathogens will evolve towards intermediate virulence, yet the necessary trade-offs invoked by this theory have rarely been demonstrated empirically. Here, the authors show that dengue virus dynamics exhibit a trade-off between transmission and clearance rates."